# Sub-micrometer refractory carbonaceous particles in the polar stratosphere

Katharina Schütze[1,2], James C. Wilson[3], Stephan Weinbruch[1], Nathalie Benker[1], Martin Ebert[1], Gebhard Günther [4], Ralf Weigel[2], Stephan Borrmann[2,5]

[1]Institut für Angewandte Geowissenschaften, Technische Universität Darmstadt, Darmstadt, Germany
[2]Institut für Physik der Atmosphäre, Johannes Gutenberg-Universität, Mainz, Germany
[3]Department of Mechanical and Materials Engineering, University of Denver, Denver, CO 80208, USA
[4]Institute for Energy and Climate Research (IEK-7), Research Center Jülich, Jülich, Germany
[5]Partikelchemie, Max-Planck-Institut für Chemie, Mainz, Germany

*Correspondence to*: Katharina Schütze (schuetze@geo.tu-darmstadt.de)

**Abstract.** Eleven particle samples collected in the polar stratosphere during SOLVE (SAGE III Ozone loss and validation experiment) from January until March 2000 were characterized in detail by high-resolution transmission and scanning electron microscopy (TEM/SEM) combined with energy-dispersive X-ray microanalysis. A total number of 4202 particles (TEM=3872; SEM=330) was analyzed from these samples which were collected mostly inside the polar vortex in the altitude range between 17.3 and 19.9 km. Particles that were volatile in the microscope beams contained ammonium sulfates and hydrogen sulfates and dominated the samples. Some particles with diameters ranging from 20 to 830 nm were refractory in the electron beams. Carbonaceous particles containing additional elements to C and O comprised from 72% to 100% of the refractory particles. The rest were internal mixtures of these materials with sulfates. The median number mixing ratio of the refractory particles, expressed in units of particles per milligram of air, was 1.1 (mg air)$^{-1}$ and varied between 0.65 and 2.3 (mg air)$^{-1}$.

Most of the refractory carbonaceous particles are completely amorphous, a few of the particles are partly ordered with a graphene sheet separation distance of $0.37 \pm 0.06$ nm (mean value $\pm$ standard deviation). Carbon and oxygen are the only detected major elements with an atomic O/C ratio of $0.11 \pm 0.07$. Minor elements observed include Si, S, Fe, Cr and Ni with the following atomic ratios relative to C: Si/C: $0.010 \pm 0.011$; S/C: $0.0007 \pm 0.0015$; Fe/C: $0.0052 \pm 0.0074$; Cr/C: $0.0012 \pm 0.0017$; Ni/C: $0.0006 \pm 0.0011$ (all mean values $\pm$ standard deviation).

High resolution element distribution images reveal that the minor elements are distributed within the carbonaceous matrix, i.e., heterogeneous inclusions are not observed. No difference in size, nanostructure and elemental composition was found between particles collected inside and outside the polar vortex.

Based on chemistry and nanostructure, aircraft exhaust, volcanic emissions and biomass burning can certainly be excluded as source. The same is true for the less probable, but globally important sources: wood burning, coal burning, diesel engines and ship emissions.

Recondensed organic matter and extraterrestrial particles, potentially originating from ablation and fragmentation remain as possible sources of the refractory carbonaceous particles studied. However, additional work is required in order to identify the sources unequivocally.

## 5 1 Introduction

The chemistry of stratospheric aerosols has been studied for more than half a century (Junge et al., 1961; Junge 1963), and it was discovered that sulfur is the main element in the particles. Junge and Manson (1961) supposed the particles to consist of ammonium sulfate, and Bigg et al. (1970) suggested sulfuric acid with varying amounts of ammonia. Rosen et al. (1971) strengthened the evidence for the material to be sulfuric acid as most of the material evaporated at the temperature expected

for this substance. According to Bigg (1975), the majority of the particles is composed of sulfuric acid with varying amounts of ammonium sulfate. A comprehensive summary of stratospheric aerosol and sulfur chemistry is given by Kremser et al. (2016).

In addition to the dominating sulfur-rich particles (sulfuric acid, sulfates), refractory particles were reported frequently. Dense,

mineral-rich particles presumably originating from the eruption of the Mt. Agung volcano were observed by Mossop (1963, 1965) using scanning electron microscopy. However, due to the lack of instrumentation, the chemistry of the particles could not be investigated. Refractory particles with diameters >1 µm were studied in more detail by Zolensky and Mackinnon (1985), and several particle groups were distinguished: chondrite, silicate, aluminum (Al), aluminum with variable amounts of other elements, iron (Fe) with or without sulfate (S), calcium (Ca)-Al silicates and "other" particles. The silicate particles were

dominant and interpreted to be volcanic, probably from the Mt. St. Helens eruption in 1980. In contrast to prior findings, a large number of refractory stratospheric particles was recognized by Zolensky et al. (1989). The particles they analyzed had diameters of >> 1 µm. The authors assumed that this increase was caused by solid rocket exhaust or the re-entry of debris associated to human space flight activity (inoperative satellites, burnt out rocket stages, tools, etc.). According to Sheridan et al. (1994) approximately 97% of all analyzed stratospheric particles were sulfuric acid. Also non-sulfate materials, soot, other

C-rich substances and crustal material were detected. Carbonaceous aerosol was found to contribute to the aerosol population at all latitudes in the stratosphere and interplanetary dust was significantly abundant above 30 km for particles ≥ 0.35 µm (Renard et al., 2008). Della Corte et al., 2013 found calcium-oxygen (CaO) -rich particles probably originating from a bolide that penetrated the Earth's atmosphere. Single particle mass spectrometry (SPMS) brought new insights into the chemistry of stratospheric particles (e.g., Murphy et al., 1998, 2007, 2013). The method is capable to measure particles in the size range of

120 nm – 3 µm (with a very low detection efficiency for particles < 220 nm; Murphy et al., 2007). According to these authors, stratospheric particles are dominated by pure sulfuric acid, sulfuric acid internally mixed with material from ablated meteoroids, and mixtures of organic-sulfate particles. A recent SEM study by Ebert et al. (2016) focused on refractory particles

in the late winter stratospheric polar vortex. The main particle groups encountered included Fe-rich, Si-rich, Ca-rich, metal mixtures and Carbon (C)/Si-rich particles.

Refractory particles in the UT/LS (Upper Troposphere / Lower Stratosphere) can act as condensation nuclei for cirrus clouds (Kojima et al., 2004; 2005; Cziczo et al., 2013) and as surfaces for heterogeneous chemical reactions in the polar stratosphere which play a significant role in polar ozone depletion (e.g., Peter, 1997; Solomon, 1999; Peter and Grooß, 2012). In addition, the particles can serve as surfaces for the heterogeneous condensation of saturated gases in the polar stratosphere (Saunders et al. 2010, 2012; Voigt et al. 2005). Due to the acidic environment, the particles can (partially) dissolve in the acidic solution droplets (binary $HNO_3$-$H_2O$ or ternary $HNO_3$-$H_2SO_4$-$H_2O$) and, thus, change their freezing properties. Therefore, the dissolved particles in a ternary $HNO_3$-$H_2SO_4$-$H_2O$ solution could have an important impact on the formation of polar stratospheric clouds (PSCs). As gaseous compounds will condense on refractory particles, they will grow, both in size and mass, which leads to a change in their sedimentation velocity (Fromm et al. 2000; Jost et al., 2004). Therefore, the gaseous compounds can both be redistributed in the stratospheric region and sediment out more quickly.

There are multiple sources which contribute to the stratospheric refractory particle load. Interplanetary dust particles are considered to be the major component of refractory material (Murphy et al., 1998; 2007; Plane, 2012). Another important source of stratospheric refractory particles are volcanic eruptions which may either eject material directly into the stratosphere (Vernier et al., 2011) or lead to particle transport through the tropical transition layer (TTL) (Mattis et al., 2010). Further potential sources of stratospheric refractory particles are high-flying aircraft (Fahey et al, 1995; Pueschel et al., 1997, Petzold et al., 1999), rockets (Newman et al., 2001), ablated material from meteorites (Hunten et al. 1980; Turco et al., 1981; Murphy et al., 1998; Cziczo et al., 2001) and all kinds of terrestrial material being lifted and entrained into the stratosphere by the Brewer-Dobson-Circulation (Holton et al., 1995, Austin and Li, 2006). As the frequency of particle emissions from the listed sources is highly variable, the individual contribution of the various sources is, in general, not quantifiable.

In summary, a variety of different refractory particle types is observed in the stratosphere. Due to the still limited number of sampling and measurement campaigns, the occurrence as well as the sources of refractory particles is not known precisely. The present paper first aims at improving the database on the observed particle groups. Second, it is attempted to infer potential sources by a detailed characterization using high-resolution transmission electron microscopy (TEM), scanning electron microscopy (SEM) and energy-dispersive X-ray microanalysis (EDX).

## 2 Experimental

### 2.1 Sampling

Stratospheric particles were sampled on board of the NASA ER-2 aircraft during the SAGE III Ozone loss and validation experiment (SOLVE), which was conducted in January-March 2000 in Kiruna (Sweden). The Multi-Sample Aerosol

Collection System (MACS) (Kojima et al., 2004), a thin-plate low-pressure impactor, was used for sampling. The particles were deposited on TEM copper (Cu) grids covered with a formvar film. MACS is designed to collect up to 23 samples per flight. The first sample is not exposed to flow and serves as a blank sample. The MACS was designed to sample and transmit all particles in the diameter range from 20 to 1000 nm to the impactor. Particles larger than approximately 20 nm are collected on the impactor in the pressure range shown in Table 1. In situ measurements of aerosol abundance were obtained with a Condensation Particle Counter (Wilson et al., 1983) simultaneously with the samples. Sampling date, flight characteristics, potential vorticity (PV), pressure altitude, pressure, potential temperature and ambient aerosol number mixing ratio (particles per milligram of air) for the analyzed samples are shown in Table 1. In situ detection of nitrogen oxides on particles (Fahey et al., 2001) was used to determine if samples were collected when the aircraft was in polar stratospheric clouds (PSCs) containing nitrogen. Measurements of $N_2O$ provide insight into the residence time of the air parcel in the stratosphere, called age of air (Wilson et al., 2008). These values are also shown in Table 1.

The meteorological conditions of the early winter (November – January) northern hemispheric polar vortex of 1999/2000 are described in detail by Manney and Sabutis, 2000. The early winter (November – January) northern hemispheric polar vortex of 1999/2000 had much lower averaged temperatures compared to any previously observed Arctic winter. The vortex was weaker than the early winter polar vortices of the previous years. It was discontinuous in the middle of December, with a large extent in the upper and small extend in the lower stratosphere. During the period of airborne measurement operations, from mid-January on, the vortex evolved to be continuous and stable until mid-March (Greenblatt et al., 2002). Jost et al. (2002) describe anomalous single mixing events occurring during that time at the potential temperature ($\Theta$) range of 350-500 K. These events are probably the result of mixing between deep vortex and extra vortex air.

The stratospheric particle samples (deposited on TEM grids) taken within the polar vortex, were packed into single plastic boxes and stored in a desiccator prior to analysis, starting in 2014. Based on the investigation of blank samples, contamination of the samples during the time of storage (e.g. by vapours from the plastic boxes) can be excluded. Furthermore, a change in particle morphology and nanostructure is not expected, since the particles found are either amorphous or show very little ordering. This conclusion is based on the fact that graphitization of carbonaceous material is an irreversible process (Diessel et al., 1978; Itaya, 1981; Pesquera and Velasco, 1988). Anyhow, it should be kept in mind that other parameters (chemical composition, mixing state) may be changed to a variable extent by aging. In total, 122 samples from 15 sampling days were collected.

## 2.2 Electron microscopy

A total of 4202 particles (3872 TEM; 330 SEM) from 11 samples were investigated by transmission and scanning electron microscopy. The samples were selected according to meteorological conditions and suitability for electron microscopy (i.e., substrate area covered by particles). Table 2 gives an overview on how many particles were investigated with which method.

The objective of this study is the detailed characterization of refractory stratospheric particles. Similar to Ebert et al., 2016, we have classified all particles that are stable (no visible morphological change) under the high vacuum conditions and electron beam excitation in the SEM and TEM as refractory.

The size, morphology, mixing state, nanostructure and chemical composition of 60 refractory particles per sample were studied by TEM using a Philips CM20 instrument (FEI, Eindhoven, The Netherlands) operated at 200 kV electron accelerating voltage. The chemical composition of the particles (all elements with an atomic number $Z \geq 5$) was determined by EDX using a Silicon-drift X-ray detector (Oxford Instruments, Oxfordshire, United Kingdom) and a measurement time of 20 seconds. Particle size and graphene sheet separation distance were analyzed by the ImageJ software (1.48v; Rasband, W. S. National Institutes of Health, USA, 1997-2016). Element distribution images were acquired with a JEOL JEM 2100F (JEOL, Tokyo, Japan) operated in Scanning Transmission Electron Microscopy (STEM) mode at an electron acceleration voltage of 200 kV. The instrument is equipped with the same type of EDX-detector as the Philips CM20 instrument.

In order to detect elements present at low abundance, additional 30 refractory particles per sample were analyzed by SEM using a Quanta 200 FEG instrument (FEI, Eindhoven, The Netherlands) equipped with an EDX detector (EDAX, Tilburg, The Netherlands) operated at 15 kV electron accelerating voltage. X-ray spectra were accumulated over fifteen minutes per particle to obtain a low detection limit. These long exposures were not feasible in the TEM due to its higher beam energy and resulting particle evaporation. Element concentrations were obtained from the X-ray count rates by applying a "standard-less" ZAF correction. Detection limits of element/carbon ratios (at% / at%) for the long-time measurements are as follows: O/C = 0.0034; Si/C = 0.0010; S/C = 0.0008; Cr/C = 0.0008; Fe/C = 0.0009 and Ni/C = 0.0011.

The particles were studied by TEM and SEM without coating.

Potential contamination of the samples was checked by investigating blanks (samples transported in the MACS but not exposed to stratospheric air flowing through the impactor orifice) for each sampling day. A few titanium (Ti) and zinc (Zn) oxide particles, as well as few pure C particles were encountered on the blank samples. They look similar to some carbonaceous particles being described as contaminants on TEM grids (Harris et al., 2001). However, these particles are different in both size and morphology compared to the carbonaceous particles observed in the impaction spot of the samples.

In order to verify that the small amounts of Fe, chromium (Cr) and nickel (Ni) detected during long-term SEM measurements are not artifacts from the substrate, five points on each substrate far away from particles were analyzed. These three elements were not detected in the measurements of the clean substrates.

Another artefact can result from scattered radiation within the SEM. This could lead to the detection of chemical elements in the vacuum chamber's housing material. To test this possibility, one sample was measured at larger sample chamber pressures ($5x10^{-3}$, 200 and 500 Pa) which result in increased scattering of beam electrons in the sample chamber. According to Stokes (2008), scattering varies between 40 – 80 % at 200 Pa and 70 – 98 % at 500 Pa. Fe, Cr, and Ni did not show increased concentrations at higher pressure, but rather their count rates decreased. The small Fe, Cr and Ni concentrations detected in individual particles were not the result of stray radiation.

The elemental composition of the particles was determined by EDX in TEM (n = 529) as well as SEM (n = 330). Due to the small size of the particles, TEM is the preferred method of analysis. As mentioned above, particles were additionally analyzed by SEM. Both measurements led to small but systematic differences in the ratios of O, Si, S, Mg (magnesium), Fe and Al to C. For example, the median O/C value is 0.236 for TEM and 0.117 for SEM (Figure S1 and S2 of the electronic supplement).

For all elements TEM-EDX yielded somewhat higher elemental ratios relative to carbon than SEM-EDX. These differences most likely result from differences in the detectors, such as thickness of detector windows, and the different acceleration voltages (15 kV in SEM versus 200 kV in TEM). However, as the differences are small (Figure S2 of the electronic supplement), our conclusions are independent of the method used. SEM-EDX data are reported here since the counting time on each particle was much longer in SEM (15 minutes) than in TEM (20 seconds) leading to higher precision as well as lower

detection limits. The much lower detection limits of SEM-EDX are important for source identification as minor elements may serve as fingerprint for several anthropogenic and natural sources. Thus, the SEM-EDX data are preferred despite the lower number of particles investigated with this technique. In addition, the chemical composition of particles could not be analyzed by TEM-EDX on two samples (labeled as G and K) due to the inappropriate position of the impaction spot on the substrate (too close to the Cu grid leading to a very high count rate for Cu).

In order to assess the mixing state of the refractory particles, additional image analysis was performed in TEM. For this purpose image series before and after TEM analysis were prepared.

All particles which showed no signs of destruction or morphological change were defined as refractory. Particles which changed under the electron beam were deemed volatile, allowing quantification of the fraction of aerosol which is volatile. In total 3316 particles were analyzed by this method.

To ensure unbiased results, the individual images as well as particles for EDX analysis were randomly taken in inner and outer areas of the impaction spot.

**2.3 Statistical analysis**

Censored boxplots show data taking into account the fraction of values below detection limit. Lower and upper quartiles appear as a box, minimum and maximum values as whiskers.

The differences in element ratios between samples collected inside and outside the vortex were tested with the generalized Wilcoxon test (Helsel, 2012) applying a significance-level of 5%. Furthermore, the differences in size, projected area diameter and element ratios between the various samples were tested with the Kruskal-Wallis rank sum test (uncensored data) and the generalized Wilcoxon test (censored data). In all individual tests, a significance level of 5 % was applied.

The detection limits for EDX data were calculated from counting statistics (background counts plus three times standard

deviation of background counts).

All statistical calculations were performed with R (version 3.3.0; R Core Team, 2016) and using the contributed package NADA (version 1.5-6; Lee, 2013).

# 3 Results

All collected particles are located within a characteristic impaction spot having a diameter of ~350 µm. As TEM bright field images show (Figure 1), volatile particles (initially deposited as droplets) cover a relatively larger area compared to refractory particles. They show high abundances of sulfur and oxygen. Sometimes, a minor nitrogen peak is also observed in the X-ray spectra. These particles are highly unstable under electron bombardment. They most likely consist of ammonium sulfate/hydrogen sulfate and formed from sulfuric/nitric acid. As the presence of sulfates in stratospheric samples is well known (e.g., Sheridan et al., 1994; Arnold et al., 1998; Murphy et al., 2007; Kremser et al., 2016), these particles are not investigated further. In addition to the sulfates, carbonaceous particles stable under electron bombardment are frequently observed either as individual particles or embedded in the sulfates (Figures 1 and 2).

As the refractory carbonaceous particles in Figure 2a have no distinct shape and surface morphology, only one TEM image is shown.

Given the size of the refractory particles and the performance of the impactor, all similar particles in the sampled air were likely delivered to the impactor and collected there. Since the amount of air drawn through the impactor is known, the atmospheric abundance of these particles can be estimated from the number of particles in the impactor sample. That number was estimated from electron micrographs sampling the impaction spot and the size if the impaction spot. The ambient number mixing ratio of the refractory carbonaceous particles varies between 0.65 (mg air)$^{-1}$ and 2.3 (mg air)$^{-1}$, with a median for all samples of 1.1 (mg air)$^{-1}$ (Table 1). When compared with CPC measurements in Table 1, the carbonaceous particles comprised a few percent of the total number of particles in the air.

*TEM Analysis of the size distribution of the particles*

The size distribution of the refractory carbonaceous particles is indicated in Figure 3. Approximately 98 % of the particles have an equivalent projected area diameter ($D_{pa}$) below 500 nm (range 20 – 830 nm). The size of the particles slightly increases with time during the campaign, sample J shows the largest median particle sizes.

*SEM Analysis of the chemical composition of the refractory carbonaceous particles*

Besides C, the refractory carbonaceous particles always contain O and Si (Figures 2, 4 and 5), and in most cases also S. The element Si may at least partly be an artifact of the substrate. The S X-ray peak in EDX-spectra originates either from sulfates internally mixed with the carbonaceous particles or from stray radiation. Please note that the heights of the individual peaks in figure 2 are not proportional to the element concentrations, but give a rough estimate of the element abundance. The elements Cr, Fe, and Ni are often found as minor component (Figure 4). These three elements exclusively occur within the carbonaceous matrix, and are not abrasion products from ice particles hitting the aerosol inlet as the metallic particles described by Murphy et al. (2004) and Martinsson et al. (2014). Furthermore, none of the samples was collected during the existence of ice particles which could potentially remove material from the impactors' inlet. During collection of samples A, B, E and G, polar

stratospheric cloud particles (PSC) containing oxides of nitrogen, $NO_y$, were abundant. As we found the refractory carbonaceous particles in all samples independent of the occurrence of $NO_y$, they are not artifacts from the removal of material from the inlet system. As the TEM substrates are made of a formvar foil predominantly consisting of C, O, and traces of Si, these three elements may – at least partly – be an artifact of the substrate. However, there are three points of evidence which clearly show that the refractory carbonaceous particles observed are not an artifact of the substrate:

1) Much higher carbon X-ray count rates were obtained when measuring particles compared to the pure substrate.
2) Graphene sheets within the particles were observed by high resolution TEM. In contrast, the substrate is always completely amorphous.
3) The refractory carbonaceous particles only occur within the impaction spot.

The refractory carbonaceous particles have a different morphology than the few carbonaceous particles found on blank samples. In addition, the carbonaceous particles encountered on blank samples are often much larger with a size of several µm. Please note that a different foil with much higher O and Si contents was used for sample K. Thus this sample was excluded from figures 4, 6 and 8. The TEM grids consist of Cu leading to strong Cu X-ray peaks in the spectra (Figure 2). Consequently, Cu is excluded from the further analysis and discussion. Mg is only present in a few particles. S is the major component of the volatile material surrounding the carbonaceous particles. Fe is found as minor element in the majority (~ 95 %), Cr in about 87 % and Ni in about 49 % of refractory particles.

The spatial distribution of minor elements within the carbonaceous particles was investigated by element distribution images in STEM (Figure 5) with a 256x256 pixel resolution as well as by measuring several points on the same particle. With both approaches it is possible to obtain highly resolved information on the spatial distribution of elements within a nanometer-scale particle. C is the most abundant element and is found in the whole particle. The elements O, Si, Cr and Fe only occur in some regions of the particles. The element Al is only detected in few particles. Due to the low number of X-ray counts, the distribution of Mg and Ni is difficult to assess. S seems to occur in the whole particle and is assumed to come from stray radiation of the surrounding sulfates. The heterogeneous element distribution was also observed in multiple point measurements (up to 20 points on one particle).

Based on elemental composition, the refractory carbonaceous particles (number n = 330) were classified into four groups (Table 3). Only few particles consist of C, O, and Si only. Many particles contain additional Cr, Fe and Ni (n = 131) or Cr and Fe (n = 125). Particles only containing additional Fe are rare (n = 22). Please note that the element S was not used for particle classification, because this element is found in most spectra, either due to scattering from surrounding volatile particles or because the refractory carbonaceous particles are embedded in sulfates.

Element ratios relative to C (at% / at%) are shown in Figure 6 for all samples. The median O/C ratio varies between 0.052 and 0.129. The median Si/C ratios vary between 0.003 and 0.012, but may be influenced by the substrate. For all other elements, the respective median ratios are generally very low (< 0.005). Sample K is not shown due to the different substrate used (with lower C and higher Si content). The differences between the various samples are for all element ratios statistically significant on the 5 % level. Most obvious, sample G has lower element/C ratios than the other samples.

*TEM analysis of the particle nanostructure*

The nanostructure was investigated by high resolution TEM. All particles are either completely amorphous (Figure 7 a, b) or show only very small regions (less than ten graphene sheets) with ordering (Figure 7 c, d). In the latter case, the graphene sheet separation distance was determined (Table 4). As these measurements are very time-consuming, only 23 particles from 5 samples were investigated. The graphene sheet separation distance varies between 0.19 and 0.60 nm, the mean values of different grains between 0.25 and 0.47 nm. This range is slightly larger than typically observed for soot (Vander Wal et al., 2010; Li et al., 2011; Liati et al., 2014; Weinbruch et al., 2016).

All samples, with the exception of one (sample D), were collected inside the polar vortex. Element ratios of the samples inside and outside the vortex are compared in Figure 8. There are no statistically significant differences on the 5% level in element ratios and particle size (Figure 3) between these two cases.

## 4. Discussion

### 4.1 Occurrence of refractory carbonaceous particles in the stratosphere

We find that all of the refractory particles are carbonaceous and typically contain minor amounts of Fe, Cr and Ni distributed within the particles. Most of the refractory carbonaceous particles are not included in or coated by sulfate. This is surprising, as the particles were sampled in air having low abundance of $N_2O$ and therefore long residence times in the stratosphere (Table 1). Therefore, one would expect that all refractory particles occurring in the polar stratosphere are covered by sulfuric or nitric acid. The low abundance of refractory particles internally mixed with sulfates contradicts expectations based on the models by Mills et al. (2005) as well as the findings of Sheridan et al. (1994) and Murphy et al. (2013) which suggest that most or all stratospheric refractory particles should be embedded in or coated with sulfuric acid. The results of our study can partly be explained by the evaporation of the sulfate component in the electron beam prior to its identification. The mixing state of the refractory carbonaceous particles may also be caused by splattering of volatile material of previously internally mixed refractory/volatile material. However, the reason for most of the refractory carbonaceous particles to be externally mixed remains open.

Refractory carbonaceous particles in the polar stratosphere were identified in several earlier studies (discussed below). Depending on the applied technique different terms were used for such particles. In the present contribution the following

nomenclature is used: All particles consisting predominantly of the element carbon are termed carbonaceous, i.e., only the chemical composition is used for classification. The term soot is used for agglomerates of primary particles (20 − 50 nm size) predominantly consisting of carbon which show a specific, onion-shell like nanostructure (Buseck et al., 2012). Black carbon (BC) is used for particles strongly absorbing light in a wide spectrum of the visible wavelength (Petzold et al., 2013) with at

least 5 $m^2g^{-1}$ at a wavelength of 550 nm (Bond et al., 2013).

Soot in the stratosphere was previously identified by scanning and transmission electron microscopy in accordance with the nomenclature outlined above (Pueschel et al., 1992 (diameter = 0.2 − 0.3 µm), 1997 (diameter ≤ 1 µm); Sheridan et al., 1994 (diameter ~ 0.3 µm); Blake and Kato, 1995 (diameter ≤0.5 µm); Strawa et al., 1999 (diameter = 0.3 − 0.4 µm); Ebert et al., 2016 (diameter ≤0.5 µm)). Carbonaceous particles (diameter ~0.1 µm) which might be soot although they were not

unequivocally identified as such, due to the lack of high resolution images, were found by Chuan and Woods (1984). Testa et al. (1990) found seven poorly graphitized carbon particles in their samples and regarded them as artifacts of carbon films from the TEM grids. As they did not provide images of these particles, they cannot be compared to our findings. However, we can exclude that our particles are substrate contaminants, since such particles did not occur on the blank samples. In addition, the refractory carbonaceous particles were only observed within the impaction spot and not on the clean substrate.

Carbonaceous particles partly containing heterogeneous metallic inclusions were found by Chen et al. (1998) (diameter = 0.1 − 2 µm). Some of these particles were called soot without providing a more precise description. Thus, it is not clear if they are similar to particles we identified as soot. According to these authors, the "soot particles" most likely stem from aircraft exhaust, as the samples were − at least partly − collected in the exhaust of an aircraft in the lower stratosphere.

Mixed carbon-sulfur particles were observed by Nguyen et al. (2008) (diameter ≤ 1 µm) at 10 km altitude between 50°N and

30°S. These particles were assumed to have formed from condensed organic matter. The differences between these particles and those found in the current study might result from differences in sampling altitudes and regions. Therefore we cannot totally exclude the particles to be different, taking into account that the particles might have evolved from condensed organic matter. However, we do not know if secondary organic particles become refractory as a result of atmospheric processes.

Stratospheric carbonaceous particles were also often detected by means of mass spectrometry (MS). For example, Murphy et al. (1998, 2013) identified carbonaceous particles, with a lower abundance in the stratosphere compared to the upper troposphere (Murphy et al., 2013; diameter = 0.3 − 0.8 µm; tropics and mid-latitudes). The same group (Murphy et al., 2007) reported the presence of a small fraction of carbonaceous particles (diameter ≤ 300 nm) within the stratosphere. As some of these particles contained potassium, they were assumed to originate from biomass burning. Particles from rocket and space

shuttle exhaust (collected in the stratosphere) were investigated by Cziczo et al. (2002) with the same instrument. In solid fuel rocket exhaust (Athena rocket and space shuttle boosters) the most frequent observed signals stem from different aluminum oxide species (often with minor amounts of Fe). About 17% of the particles (Athena rocket) were classified as carbonaceous as compared to 1% from space shuttle exhaust. In principle, the carbonaceous particles found by MS could be similar to the

carbonaceous particles of the present study. Since MS does not yield images of the particle morphology or information on the nanostructure, no definite conclusion can be drawn.

Stratospheric carbonaceous particles were also identified by optical measurements. For example, a single particle soot photometer (SP2) was applied by Baumgardner et al. (2004) (diameter = 0.15 − 1 µm) on board of an aircraft in order to identify ozone loss processes in the polar vortex. In total, 60% of the light absorbing particles incandesce at temperatures above 3500 K and are, thus, interpreted as BC. According to Baumgardner et al. (2004) these particles originate from tropospheric sources rather than aircraft emissions. The same technique was applied by Schwarz et al. (2006) (diameter = 0.15 − 0.7 µm) to identify BC in midlatitudes from the boundary layer to the lower stratosphere. Only ≤1% of the particles was classified as BC and no potential source was specified.

Local enhancements of carbonaceous material at altitudes around 25 km were also deduced from simultaneous measurements of a spectrometer on board a satellite as well as from radiance and particle counter data (Renard et al., 2008) (diameter = 0.35 − 2 µm) obtained on a stratospheric balloon. This material was thought to be injected into the stratosphere by the pyro-convective effect (i.e., fire-started or fire added convection). Some of the particles with submicron size were supposed to originate from vaporized interplanetary material (Renard et al., 2008). Due to the lack of information on morphology, chemistry and microstructure of the particles, a direct comparison of the carbonaceous material deduced from optical measurements with the particles encountered in the present study is not possible.

In the present study, only carbonaceous particles and sulfates were observed similar to previous findings (Pueschel et al., 1992; Blake and Kato, 1995; Strawa et al., 1999; Nguyen et al., 2008). There are, however, several previous publications which describe the presence of a variety of other refractory particle groups in addition to carbonaceous particles. These additional particle groups include metallic particles (Chuan and Woods, 1984; Sheridan et al., 1994; Chen et al., 1998; Baumgardner et al., 2004; Ebert et al., 2016), meteoritic particles (Murphy et al., 1998, 2007, 2013; Renard et al., 2008, Ebert et al., 2016), silicates (Testa et al., 1990; Ebert et al., 2016), crustal-type particles (Sheridan et al., 1994; Chen et al., 1998), as well as Ca-bearing particles (Della Corte et al., 2013; Ebert et al., 2016).

In summary, the sole occurrence of refractory carbonaceous particles and sulfates in stratospheric samples was reported in previous literature but seems to be uncommon. The median number mixing ratio (1.1 mg air$^{-1}$) of carbonaceous particles is smaller by an order of magnitude than the abundance of non-volatile particles reported by, e.g., Weigel et al. (2014) for measurements in the winter stratospheric polar vortex in 2003, 2010 and 2011. The method described by Weigel et al. involves exposure of particles to a temperature >250 °C and determination (with a CPC) of the number of particles that did not evaporate to sizes below the detection limit of the CPC. They concluded that up to 80% of the particles present were non-volatile by this criterion. Following our definition only a few percent of the SOLVE particles are non-volatile in the electron microscope. This discrepancy may be caused by the different definitions of a non-volatile particle.

## 4.2 Potential Sources

The most likely sources of refractory carbonaceous particles found in the current study include aircraft emissions, extraterrestrial sources, rocket exhaust and explosive volcanic eruptions as these sources emit material directly into the stratosphere. In addition, biomass burning can be considered as a possible source, as it was shown by several authors that large fires have sufficient energy to inject particles into the lower stratosphere (Fromm et al., 2000, 2006, Siebert et al., 2000;, Fromm and Servranckx, 2003; Jost et al., 2004; Siddaway et al., 2011). Domestic wood burning, coal combustion, diesel engines and ship exhaust are not expected to significantly contribute to the stratospheric particle load. However, as these sources emit large amounts of carbonaceous material on a global scale (Bond et al., 2004, Gaffney and Marley, 2005, Corbett and Koehler, 2003; Lauer et al., 2007), they will be discussed shortly. For example Thornberry et al., 2010 show that tropical tropospheric particles contain high amounts of thermostable carbonaceous material, and it is possible that these particles become mixed within stratospheric air. Although the vertical exchange may be less effective than the direct injection processes, however, it is conceivable that fractions of carbonaceous aerosol material released in the troposphere is vertically transported into the stratosphere by processes such as tropical convection (and lifted further via the Brewer-Dobson circulation), cyclogeneses, warm conveyor belts, tropopause folds and/or isentropic transport.

Most particle groups discussed in the following were collected close to their emission source. We are aware of the fact, that particles collected in the polar stratosphere may in principle change their properties during their atmospheric lifetime. However, ordering of carbonaceous material is an irreversible process leading always to a higher degree of ordering (Diessel et al., 1978; Itaya, 1981; Pesquera and Velasco, 1988). As most of the particles analyzed show no or only very little ordering, it is assumed that the particles did not change their nanostructure during their atmospheric lifetime. On the other hand, several electron microscopy studies describe soot particles in the stratosphere (Pueschel et al., 1992, 1997; Sheridan et al., 1994; Strawa et al., 1999; Ebert et al., 2016). Thus, it can be expected that soot particles - once injected into the stratosphere – do not change their typical nanostructure under stratospheric conditions.

*Aircraft exhaust*

High flying aircraft can contribute significantly to the stratospheric aerosol burden. Soot is described as the main particulate exhaust component (Twohy and Gandrud, 1998; Popovitcheva et al., 2000; Smekens et al., 2005; Vander Wal et al., 2010; Liati et al., 2014). The observed soot consisted of primary particles 10 - 50 nm in diameter which formed aggregates with diameters of up to approximately 450 nm (Twohy and Gandrud, 1998; Popovitcheva et al, 2000; Smekens et al., 2005; Liati et al., 2014). The nanostructure of the primary particles varied from an onion-shell structure with partial ordering (Popovitcheva et al., 2000; Vander Wal et al., 2010) to completely amorphous (Twohy and Gandrud, 1998). The mean graphene sheet separation distance of the partly ordered primary particles varied between 0.36 and 0.41 nm (Vander Wal et al., 2010; Liati et al., 2014). An atomic O/C ratio of 0.32 was reported by Vander Wal et al. (2010). The elements S, Na (sodium), N (nitrogen), Zn (zinc), Ba (barium), Cl (chlorine), K (potassium), Fe and Si were detected in minor concentrations

(Vander Wal et al., 2010; Mazaheri et al., 2013). In addition to soot agglomerates, single carbon spheres were found by Mazaheri et al. (2013) in aircraft exhaust. The particles have diameters between 5-100 nm with a median of 35.4 nm. Besides C, minor O, S, Cl, K, Fe and Si were detected by TEM-EDX and Proton Induced X-ray Emission (PIXE) analysis. Aircraft exhaust is excluded as source of the carbonaceous particles encountered in the present study, as we did not observe soot agglomerates. In addition, the chemistry and morphology (basically rounded shapes) of the single carbon particles described by Mazaheri et al. (2013) are different from our particles.

*Extraterrestrial particles*

With an input of 5-270 tons per day (Plane, 2012), extraterrestrial material is expected to be the major source of refractory stratospheric particles (Murphy et al., 1998; 2007). Carbonaceous material is observed in chondrites (dominant meteorite fraction) as well as in interplanetary dust particles (IDPs).

In carbonaceous chondrites, a variety of different carbonaceous constituents is described in previous literature. For example, nanometer-sized carbon-rich flakes, spheres and tubes as well as hollow carbon-rich nanospheres were found (Garvie and Buseck, 2004; Garvie, 2006; Garvie et al., 2008). Most carbonaceous nanospheres are amorphous. Besides C, the only other elements detected are S, N and O. Carbonaceous material in carbonaceous chondrites was also investigated by Aoki and Akai (2008). Different morphologies like "ribbon-film–like carbonaceous material", "spherical carbonaceous globules", "concentric-sphere type carbon material", and "featureless carbon material" were observed. They also describe the occurrence of minor amounts of Cl in many of these particles, an element never observed in our study. Neither morphology nor shape or chemical composition of the particles described above matches the refractory carbonaceous particles encountered in the present study. Carbon nanoparticles with diameters between 2-10 nm were observed in carbonaceous material isolated from the Allende carbonaceous chondrite (Harris et al., 2000). The particles had either a single outer wall or were surrounded by multiple walls. However, as the particles were mobile under the electron beam, no photographs are shown. A comparison to our particles is thus impossible. Since the authors did not find any other elements except C, most of our particles differ in chemical composition as they contain Fe, Cr, and/or Ni as minor elements.

Carbonaceous material was also found in IDPs (Matrajt et al., 2012). Some of the material observed is similar in size, morphology and nanostructure to our particles. However, in Matrajt et al. (2012) minor elements were not investigated and this parameter cannot be compared.

In summary, the carbonaceous components observed in chondrites differ in chemical composition from most of our particles. Carbonaceous material contained in IDPs cannot be excluded as source of the refractory particles encountered in the present study, as the minor element concentration of the former is not known. However, if we expect extraterrestrial material to be the major source of our particles, we would also expect to find Mg-rich silicates in our samples, which was not the case. Furthermore, the occurrence of Fe, Cr and Ni as minor elements in our refractory carbonaceous particles is regarded as hint for an anthropogenic origin. This interpretation is supported by the fact that the observed average atomic ratios of these three

elements (Cr/Fe = 0.249, Ni/Fe = 0.167, Cr/Ni = 1.145) are significantly higher than the cosmic element ratios (Cr/Fe = 0.015, Ni/Fe = 0.056, Cr/Ni = 0.278; Palme and Jones, 2005).

The chemical composition of extraterrestrial material may be strongly fractionated by frictional heating during atmospheric entry (e.g., Carrillo-Sánchez et al., 2016; Gómez- Martin et al., 2017). The processes taking place during atmospheric entry
include ablation by sputtering and thermal evaporation as well as fragmentation. Meteorite ablation particles usually occur as iron, glass or silicate spheres (e.g., Blanchard et al., 1980; Murrell et al., 1980). Submicrometer refractory carbonaceous particles resulting from meteoric ablation and fragmentation have - to the best of our knowledge - not been described in previous literature. However, it is conceivable that such particles originate from carbonaceous material contained in meteorites or interplanetary dust particles. The observed non-chondritic ratios of the minor elements Fe, Cr, Ni are not a strong argument
against such an origin, as the volatility of these elements depends on the minerals in which they are contained. Most of extraterrestrial Fe occurs as metal, silicate or oxide, most of Ni as metal (Papike, 1998). Cr may occur as oxide, sulphide or nitride as well as a minor component in metal and silicates (Bunch and Olsen, 1975). Depending on the relative abundance of the different mineral phases, substantial fractionation during evaporation can be expected (see also Floss et al., 1996). In summary, meteoric ablation and fragmentation particles are a possible source of the particles encountered in the present study.

*Rocket Exhaust*

Rocket exhaust is also a possible source of stratospheric particles. However, literature on particles emitted by rockets is sparse (e.g. Zolensky et al., 1989), and there are – to the best of our knowledge – no studies available on carbonaceous particles by electron microscopy. According to Ross and Sheaffer (2014), five different propellant types which use a combination of
different oxidizers and fuels are in use: $O_2$/kerosene, $O_2/H_2$, $NH_4ClO_4$/Al, $N_2H_4/N_2O_4$ and $N_2O$/solid hydrocarbons. Solid rocket motors (SRM) emit characteristic $Al_2O_3$ spheres (Strand et al., 1981; Zolensky et al., 1989; Cofer III et al., 1991) and can, thus, be excluded as source of the carbonaceous particles encountered in the present study. Hydrocarbon-fired rockets are powered by kerosene or syntin and can be expected to emit soot. For example, soot particles most likely emitted by a Russian Soyuz booster rocket were found in an aerosol cloud at 20 km (Newman et al., 2001). The soot particles contribute
approximately 1/4 to the total particle number, the rest were volatile sulfate particles. The occurrence of carbonaceous material from rocket exhaust was also reported by Cziczo et al. (2002). In the exhaust of an Athena II rocket, the carbonaceous fraction of material was found to be 17% by number (Cziczo et al., 2002). As the refractory carbonaceous particles observed by us are not soot, their origin from rocket exhaust is unlikely. However, as carbonaceous rocket exhaust particles were not investigated previously by electron microscopy this source cannot be excluded.

*Volcanic emissions*

Volcanic eruptions are generally not considered to emit carbonaceous material. However, carbonaceous material was found in samples from the Kasatochi (Alaska, 2008), Sarychev (Russia, 2009) as well as Eyafjallajökull (Island, 2010) eruptions (Martinsson et al., 2009; Schmale et al.; 2010; Andersson et al., 2013). The carbonaceous material most probably originates

from air entrained into the volcanic cloud. The carbonaceous mass fraction in plumes from two volcanoes in Ecuador and Columbia was found to vary between 19-38% (measured by MS; Carn et al., 2011). Sulfur-carbonaceous mixed particles occurred predominantly in the size range below 0.9 µm. In 1999, the year before the campaign, there were four volcanic eruptions with a VEI ≥ 4 (Volcanic Explosivity Index): Soufrière Hills (West Indies) in January, Shiveluch (Russia) in August as well as Guagua Pichincha (Ecuador) and Tungurahua (Ecuador) in October (NOAA, 2017). Still, volcanism seems an unlikely source for our samples, as volcanic eruptions will emit large amounts of silicate particles, which we did not observe.

*Biomass burning*

A further source of carbonaceous particles in the stratosphere is biomass burning (BB). Particles originating from BB can be lifted to the stratosphere by either the tropical upper tropospheric upwelling or the pyroconvective effect (Fromm et al., 2000; Jost et al., 2004). Three different types of carbonaceous BB particles were described in previous literature (e.g., Pósfai et al., 2003; 2004; Kis et al., 2006; Li and Shao, 2009): organic particles with inorganic inclusions, tar ball particles and soot. Organic particles (no specific morphology) contain C and minor O, and are stable under the electron beam. They do not show the typical microstructure of soot (see below). The inorganic inclusions mostly consist of KCl and $K_2SO_4$. Tar balls have a typical spherical shape and mainly consist of C and O with minor K, S, Cl and Si contents. Soot consists of chain-like agglomerates of primary particles (10 – 100 nm) with a typical onion shell microstructure (graphene sheet separation distance between 0.133 and 0.137 nm). As our particles do not show the characteristics of all types of carbonaceous BB particles described above, this source can be excluded.

The most probable potential sources for carbonaceous particles were already discussed above. For these sources the transport mechanisms into the stratosphere are well known. There are further strong tropospheric sources for carbonaceous particles, which predominately emit particles at ground level. An effective transport to the stratosphere of these particles is unlikely. Still, they will be discussed briefly, as they are – on a global scale – major sources of carbonaceous material in the lower atmosphere.

*Wood burning*

Soot is a major component emitted by wood burning. Similar to biomass burning, soot from wood burning consists of agglomerates of spherical primary particles (20 – 80 nm diameter) with an onion-shell nanostructure (Kocbach et al., 2005; Torvela et al., 2014, Tumolva et al., 2010). Some primary particles are amorphous (Tumolva et al., 2010). The particles may have a surface coating which is volatile under electron bombardment (Torvela et al., 2014). Carbon is the major element of wood burning soot, O, Na, Si, S, Cl, K, and Ca occur as minor elements (Kocbach et al., 2005; Tumolva et al., 2010). In addition to soot, particulate organic matter (POM) was found in wood burning (Torvela et al., 2014). The POM particles, sometimes described as tar balls are 30 - 600 nm in diameter. Judged from the properties described above, both soot and POM

are different to the particles observed in our study. Thus, wood burning can be excluded as a source for the refractory carbonaceous stratospheric particles.

*Coal burning*

5 A variety of different carbonaceous particles was observed during coal burning. Most carbonaceous particles are soot particles, i.e., fractal-like agglomerates (0.1 – 1 µm size) consisting of 10 – 50 nm diameter primary particles. The primary particles show a characteristic onion-shell structure (Chen et al., 2004, 2005). Some of the soot agglomerates have inorganic inclusions or inorganic particles on their surface, containing Mg, Ca, Sr (strontium), Ba (barium) and Na (Chen et al., 2005). Char particles associated with ultrafine titanium oxide particles were also found (Chen et al., 2005). Sometimes ultrafine Ti, Al, Fe and Ca
10 particles were embedded in large char particles. In addition, graphitic fiber structures that are either straight or curved were encountered. Two of the three types of carbonaceous particles described for coal combustion (soot agglomerates and graphitic fibers) are certainly different from our particles. As images of char particles were not provided by Chen et al. (2005) we cannot compare their results to the refractory carbonaceous particles of the present study. However, we regard coal combustion as an implausible source.

*Diesel engines*

Diesel engines are another important source of carbonaceous particles and above all soot particles. Again, soot agglomerates consist of spherical primary particles (5 – 50 nm diameter) with onion-shell nanostructure (e.g., Tumolva et al., 2010; Wentzel et al., 2010; Li et al. , 2011; Song, 2004; Weinbruch et al., 2016). The graphene sheet separation distance varies between 0.31
20 – 0.48 nm; (Vander Wal et al., 2010; Li et al., 2011; Weinbruch et al., 2016), and the intensity O/C ratio between 0.049 – 0.079 (Weinbruch et al., 2016). Compared to our particles, the chemical composition, morphology and primary particle size of diesel exhaust particles are significantly different. Thus, we can certainly exclude this source.

*Ship emissions*

25 Ship emissions also contain soot agglomerates (Popovicheva et al., 2012; Lieke et al., 2013). Their chemical composition is dominated by C and O, with small amounts (< 1 wt %) of additional elements; e.g. V, S, Cl, Ca and Si (Popovicheva et al., 2012). Furthermore, spherical char particles with diameters of 0.2 – 1 µm are found to be characteristic for ship emissions (Popovicheva et al., 2012). The morphology of these particles shown in SEM photomicrographs is different to the particles found in our study. Lieke et al., 2013 found amorphous carbonaceous material filling cavities of larger soot aggregates. The
30 characteristics described for ship exhaust particles are significantly different to the particles found in our study, leading to the conclusion that that ship exhaust can be excluded as the source.

## 5. Summary

The major finding of the present study is that the refractory component consists of carbonaceous particles only, with a number mixing ratio of 1.1 $(mg\ air)^{-1}$ (median for all samples). Most carbonaceous particles are not internally mixed with or coated by sulfates. The particles were sampled in air having low abundance of $N_2O$ and therefore long residence times in the stratosphere. Thus, one would expect them to be covered with condensed sulfuric acid resulting from the oxidation of COS (Wilson et al., 2008). The reason for this discrepancy is not known.

As major elements only C and O were detected. Most of the carbonaceous particles show small and variable amounts of Fe, Cr and Ni. These minor elements are distributed in the carbonaceous matrix, i.e., they do not occur as heterogeneous inclusions. Most carbonaceous particles are completely amorphous.

The exact source of the refractory carbonaceous particles remains unclear and can only be confined by exclusion. Based on the investigated physical properties and chemical composition of the particles, aircraft exhaust, volcanic emissions and biomass burning can be certainly excluded as source. The same is true for the even more unlikely sources wood burning, coal burning, diesel engines and ship emissions. It is expected that exhaust of rockets powered by kerosene or other hydrocarbons emit soot, but due to the lack of available electron microscopy studies of these emissions, rocket exhaust cannot be excluded as a possible source of the refractory carbonaceous particles found. Carbonaceous material from IDPs and extraterrestrial particles, likely originating from meteoric ablation and fragmentation remain as the most probable source for the particles encountered in the current study.

**Data availability.** The data set is available for the community and can be accessed by request to Katharina Schütze (schuetze@geo.tu-darmstadt.de) of the Technische Universität Darmstadt.

**Author contributions.** Katharina Schütze analyzed the samples, interpreted the data and prepared the manuscript. Nathalie Benker supported the TEM analysis. Martin Ebert came up with suggestions for and supported the SEM analysis. Stephan Weinbruch contributed to the preparation of the manuscript as well as interpretation and data analysis. Stephan Borrmann and Ralf Weigel supported the scope of the analysis as well as interpretation of data and writing. James C. Wilson provided the samples as well as any necessary data regarding the sampling campaign, contributed to the data analysis, interpretation and writing. Gebhard Günther conducted calculations on the meteorological situation of the collected samples.

**Competing interests.** All authors declare that they have no conflict of interest.

**Acknowledgements.** We thank the two anonymous reviewers and Alexander D. James for their helpful comments that considerably improved the manuscript. Our research was funded by the European Research Council under FP7 (FP/2007-2013)/ERC grant agreement no. 321040 (EXCATRO). This work was partly supported by the project ROMIC-SPITFIRE

sponsored by the Federal Ministry of Education and Research (FKZ 01LG1205D) and by STRATOCLIM sponsored by the European Union Seventh Framework Program (FP7), project reference 603557.

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

**Table 1:** Sampling conditions

| Table 1: Sampling conditions date (y/m/d/no) | sample In Vortex Out of Vortex | characteristics of flight* / Nitrogen Containing PSCs sampled | PV at Θ [PVU] | altitude [km] pressure [hPa] potential temperature Θ [K] # | CPC Aerosol Number Mixing Ratio, Number per milligram of air (mg air)$^{-1}$ | N$_2$O, Age of Air |
|---|---|---|---|---|---|---|
| 2000-01-20_07 | A In Vortex | PSC survey§/ PSCs sampled | 26.2 | 19.7/ 57/ 433 | Not available | 165 ppbv 4.3 years |
| 2000-01-20_11 | B In Vortex | PSC survey§/ PSCs sampled | 26.6 | 19.4/ 60/ 431 | Not available | 171 ppbv 4.2 years |
| 2000-01-23_18 | C In Vortex | vortex, sunrise / PSCs  sampled | 31.3 | 19.8/ 56/ 438 | 87 | 153 ppbv 4.5 years |
| 2000-01-27_15 | D Out of Vortex | edge survey / No PSCs | 24.7 | 19.9/ 56/ 448 | 72 | 227 ppbv 3.6 years |
| 2000-01-31_18 | E In Vortex | vortex survey/ PSCs sampled | 32.0 | 19.7/ 59/ 437 | 88 | 171 ppbv 4.2 years |
| 2000-02-02_19 | F In Vortex | vortex survey/ PSC Unknown/ | 26.5 | 18.6/ 68/ 425 | 86 | 167 ppbv 4.3 years |
| 2000-02-03_15 | G In Vortex | multiple level/ PSCs sampled/ | 18.2 | 17.4/ 83/ 400 | 65 | 209 ppbv 3.7 years |
| 2000-02-26_12 | H In Vortex | vortex survey/ No PSCs | 30.3 | 19.1/ 62/ 430 | 87 | 143 ppbv 4.6 years |
| 2000-02-26_14 | I In Vortex | vortex survey/ No PSC s | 30.4 | 17.3/ 64/ 431 | 90 | 137 ppbv 4.7 years |
| 2000-03-05_19 | J In Vortex | vortex survey PSC unknown | 27.7 | 19.2/ 64/ 424 | 63 | 150 ppbv 4.5 years |
| 2000-03-11_19 | K In vortex | vortex edge PSCs unknown | 29.6 | 18.7/ 66/ 430 | 89 | 136 ppbv 4.7 years |

*according to Newman et al. (2002), #during sampling, § no particles out of PSCs were analyzed; PV= Potential Vorticity,

PVU= Potential Vorticity Unit [$10^{-6}$ K m$^2$ kg$^{-1}$ s$^{-1}$]

**Table 2**: Overview over particle parameters investigated.

| Information | Number of particles investigated | Applied in Figure/Table | Method |
|---|---|---|---|
| Size, morphology, chemical composition | 529 | Figure 3 | TEM |
| Mixing state, morphology | 3316 | --- | |
| Nanostructure | 23 | Table 4, Figure 7 | |
| Distribution of elements within the particles | 4 | Figure 5 | STEM |
| Size, morphology, chemical composition | 330 | Figure 3, Figure 4, Figure 6, Figure 8 | SEM |
| Sum of total particles | 4202 | | |

**Table 3:** Absolute number of refractory particles as function of particle group.

| Particle group | sample | | | | | | | | | | |
|---|---|---|---|---|---|---|---|---|---|---|---|
| | A | B | C | D | E | F | G | H | I | J | K |
| **C, O, Si** | 4 | 8 | 3 | 2 | 4 | 8 | 11 | 3 | 4 | 3 | 2 |
| **+ Cr, Fe, Ni** | 15 | 6 | 15 | 10 | 14 | 9 | 3 | 12 | 14 | 21 | 12 |
| **+ Cr, Fe** | 10 | 13 | 12 | 13 | 12 | 9 | 11 | 13 | 12 | 4 | 16 |
| **+ Fe** | 1 | 3 | 0 | 5 | 0 | 4 | 5 | 2 | 0 | 2 | 0 |

**Table 4:** Graphene sheet separation distance

| Particle[#] | Separation distance [nm] | | |
|---|---|---|---|
| | mean value | minimum - maximum | n |
| A-1 | 0.38 | 0.35 – 0.39 | 3 |
| B-1 | 0.35 | 0.23 – 0.45 | 10 |
| C-1 | 0.35 | 0.33 – 0.37 | 5 |
| C-2 | 0.39 | 0.35 – 0.45 | 4 |
| C-3 | 0.41 | 0.37 – 0.48 | 5 |
| C-4 | 0.37 | 0.32 – 0.42 | 4 |
| C-5 | 0.38 | 0.34 – 0.40 | 6 |
| C-6 | 0.39 | 0.33 – 0.47 | 5 |
| C-7 | 0.38 | 0.35 – 0.40 | 5 |
| G-1 | 0.42 | 0.34 – 0.51 | 14 |
| G-2 | 0.47 | 0.37 – 0.60 | 18 |
| G-3 | 0.42 | 0.38 – 0.49 | 15 |
| G-4 | 0.43 | 0.40 – 0.53 | 35 |
| G-5 | 0.43 | 0.36 – 0.51 | 20 |
| G-6 | 0.29 | 0.19 – 0.32 | 25 |
| G-7 | 0.43 | 0.38 – 0.51 | 20 |
| I-1 | 0.34 | 0.31 – 0.39 | 4 |
| I-2 | 0.27 | 0.24 – 0.29 | 4 |
| I-3 | 0.27 | 0.25 – 0.28 | 3 |
| I-4 | 0.25 | 0.23 – 0.26 | 3 |
| I-5 | 0.46 | 0.44 – 0.48 | 3 |
| I-6 | 0.30 | 0.28 – 0.33 | 4 |
| I-7 | 0.33 | 0.32 – 0.34 | 3 |

[#]sample-particle

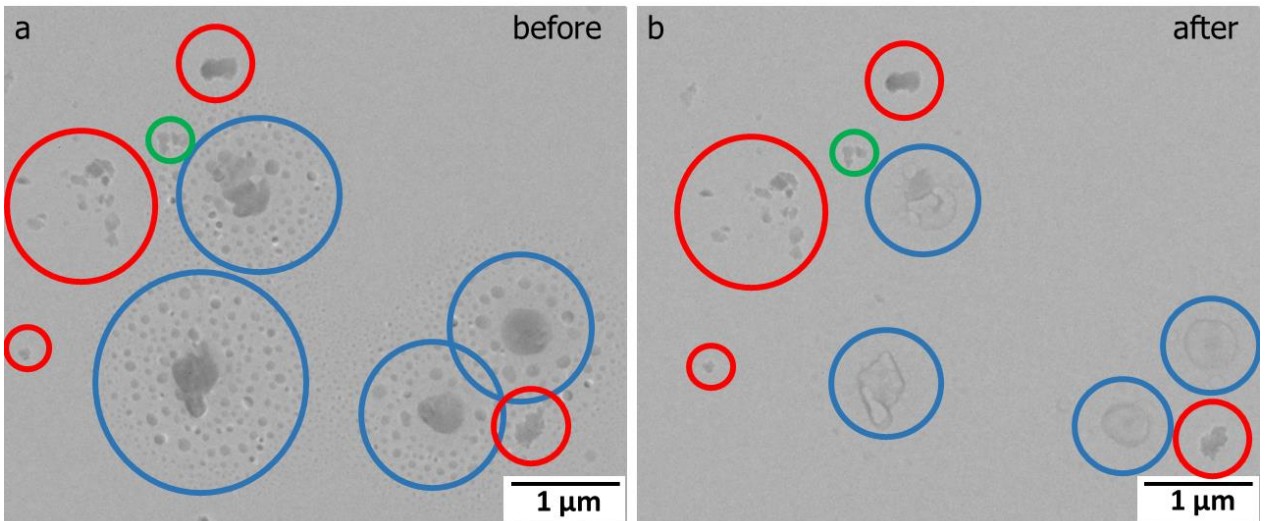

**Figure 1:** TEM bright field image of a typical sample (sample I; 17.3 km altitude), before (a) and after evaporation (b). Particles evaporating under electron bombardment are marked with blue circles. They consist of sulfates/hydrogen sulfates. Red circles indicate stable carbonaceous particles. Green circles show refractory carbonaceous particle internally mixed with volatile sulfates/hydrogen sulfates.

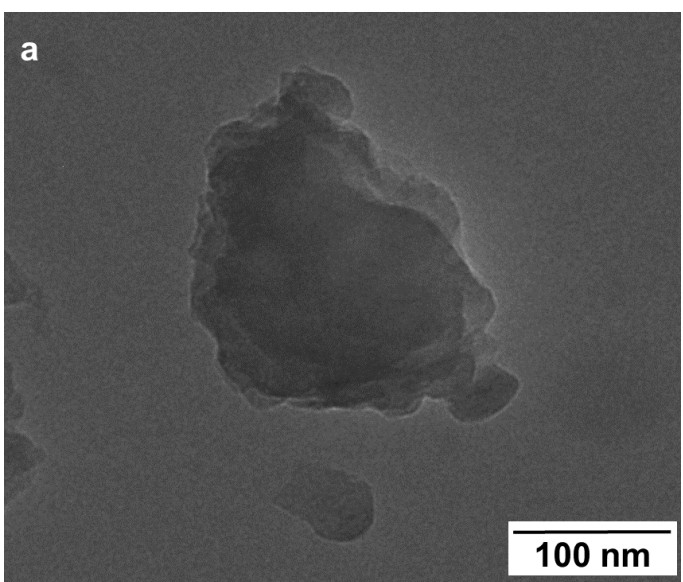

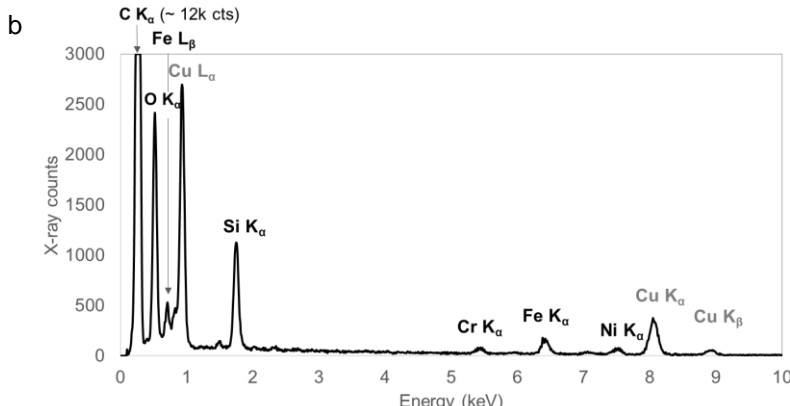

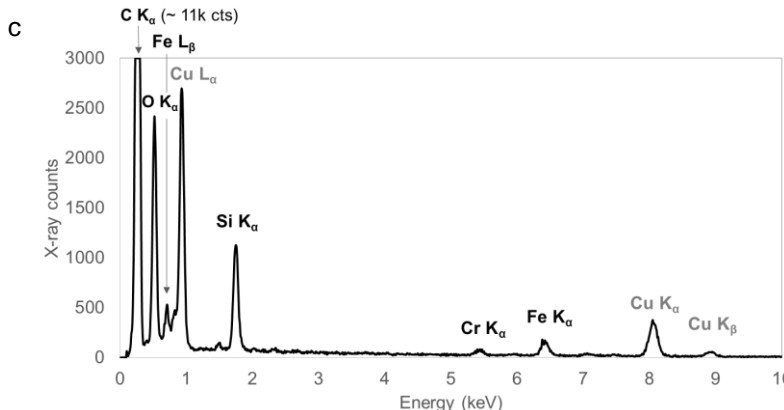

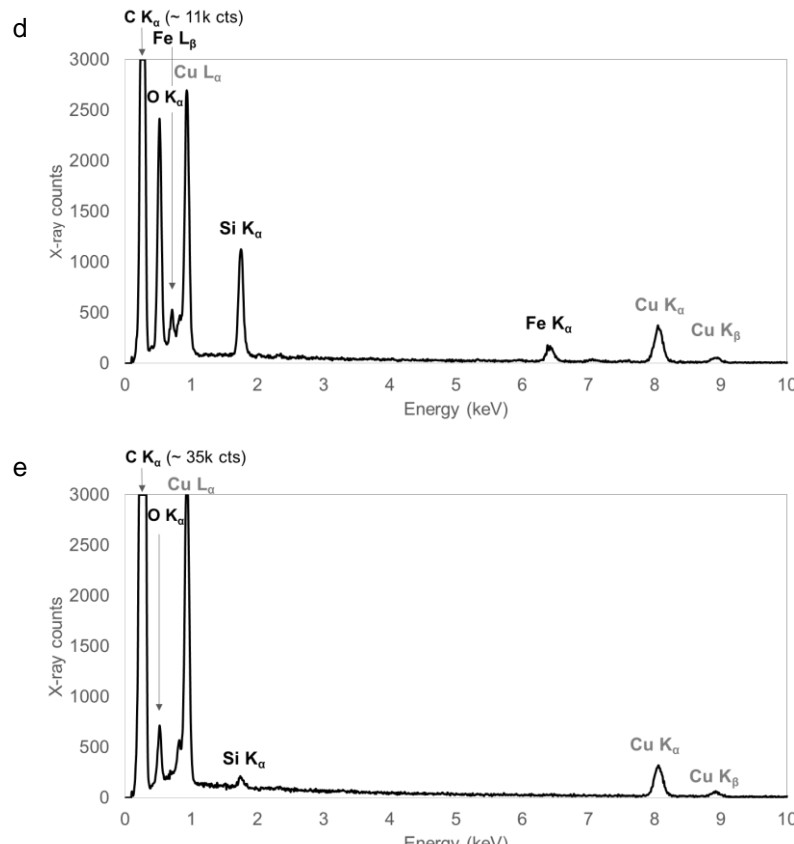

**Figure 2:** TEM bright field image (a) of a typical refractory carbonaceous particle from sample H (19.1 km altitude). The image is representative for all refractory carbonaceous particles. The morphology is not depending on chemical composition, size, morphology or nanostructure. Energy-dispersive X-ray spectra of (b) a typical refractory carbonaceous particle with Fe, Cr and Ni, (c) Fe and Cr, (d) Fe and (e) without any other minor constitute. The particle predominantly consists of C and O. Minor amounts of Si are always present and may partly be an artifact of the substrate. Cu is an artifact from the TEM grid. $K_\alpha$ and $K_\beta$ as well as $L_\alpha$ and $L_\beta$ denote different X-ray peaks emitted from the same element.

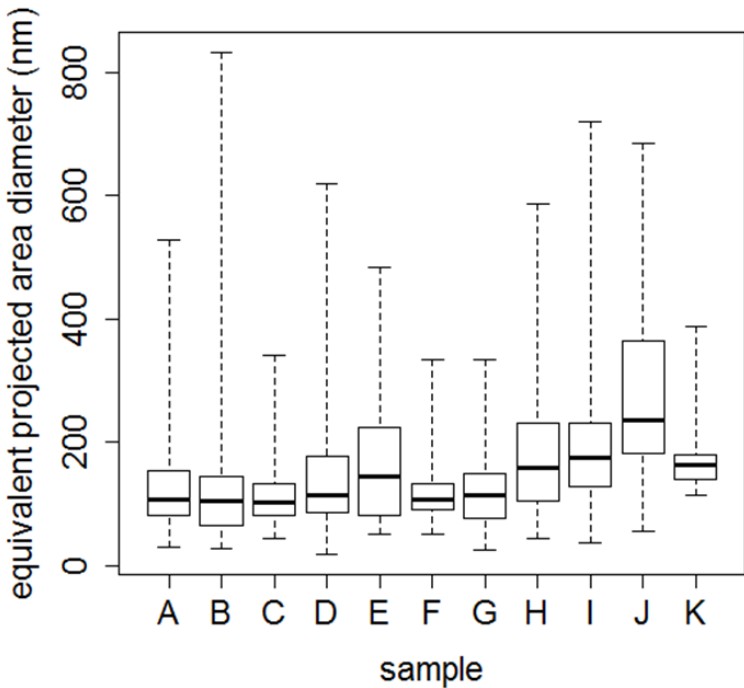

**Figure 3:** Boxplot of particle size (equivalent projected area diameter $D_{pa}$). Lower and upper quartiles appear as a box, minimum and maximum values as whiskers.

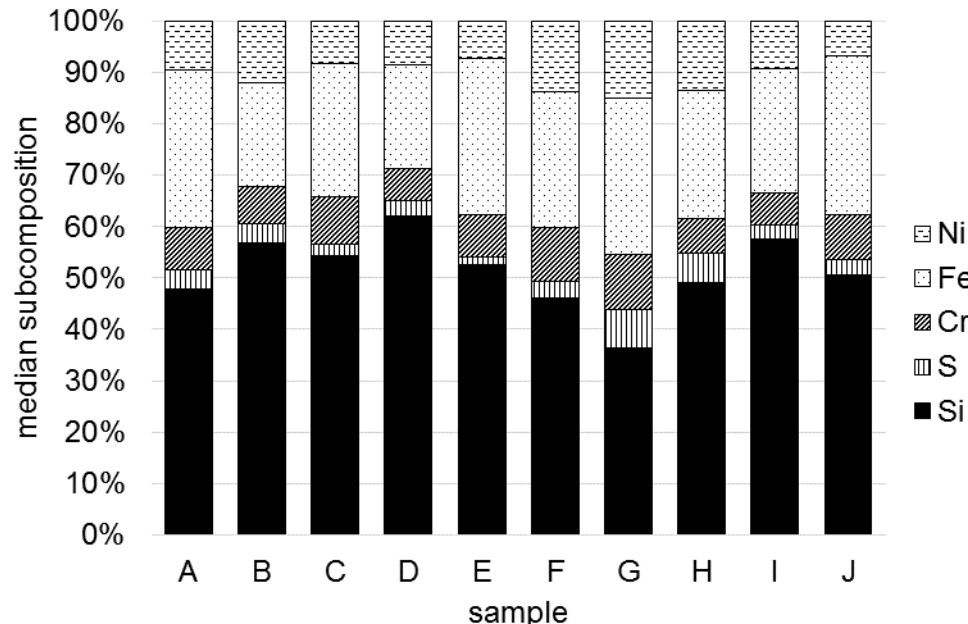

**Figure 4:** Median chemical subcomposition (atom %, without C and O) of refractory carbonaceous particles determined by SEM-EDX (30 particles per sample). Sample K was excluded from the figure due to a different substrate with higher Si content.

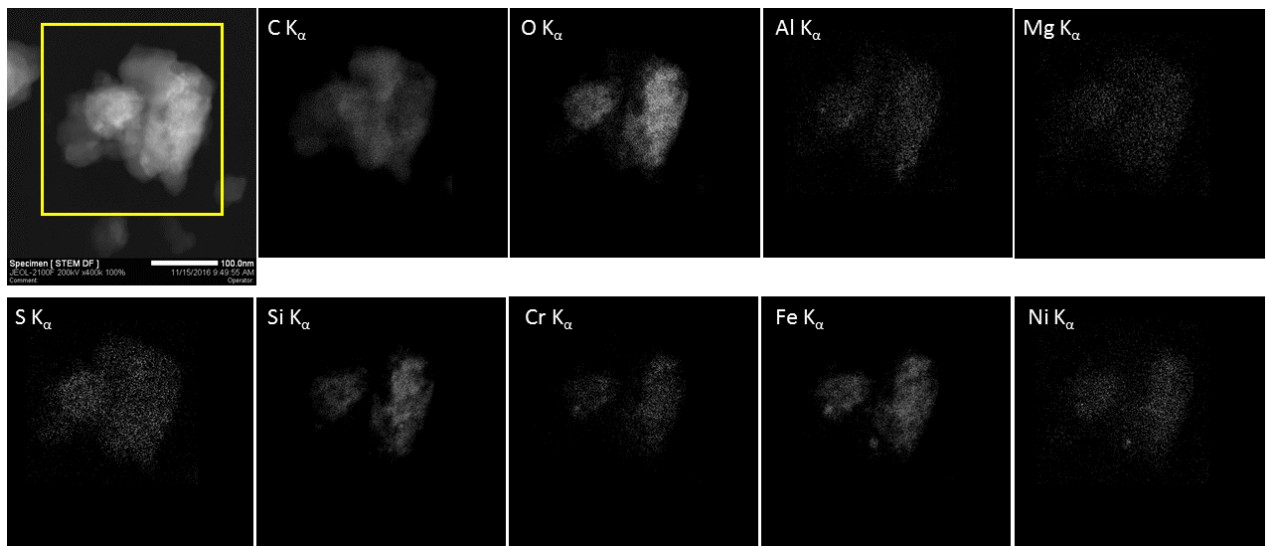

**Figure 5**: STEM image (upper left) and element distribution images of a refractory carbonaceous particle from sample C (19.8 km altitude).

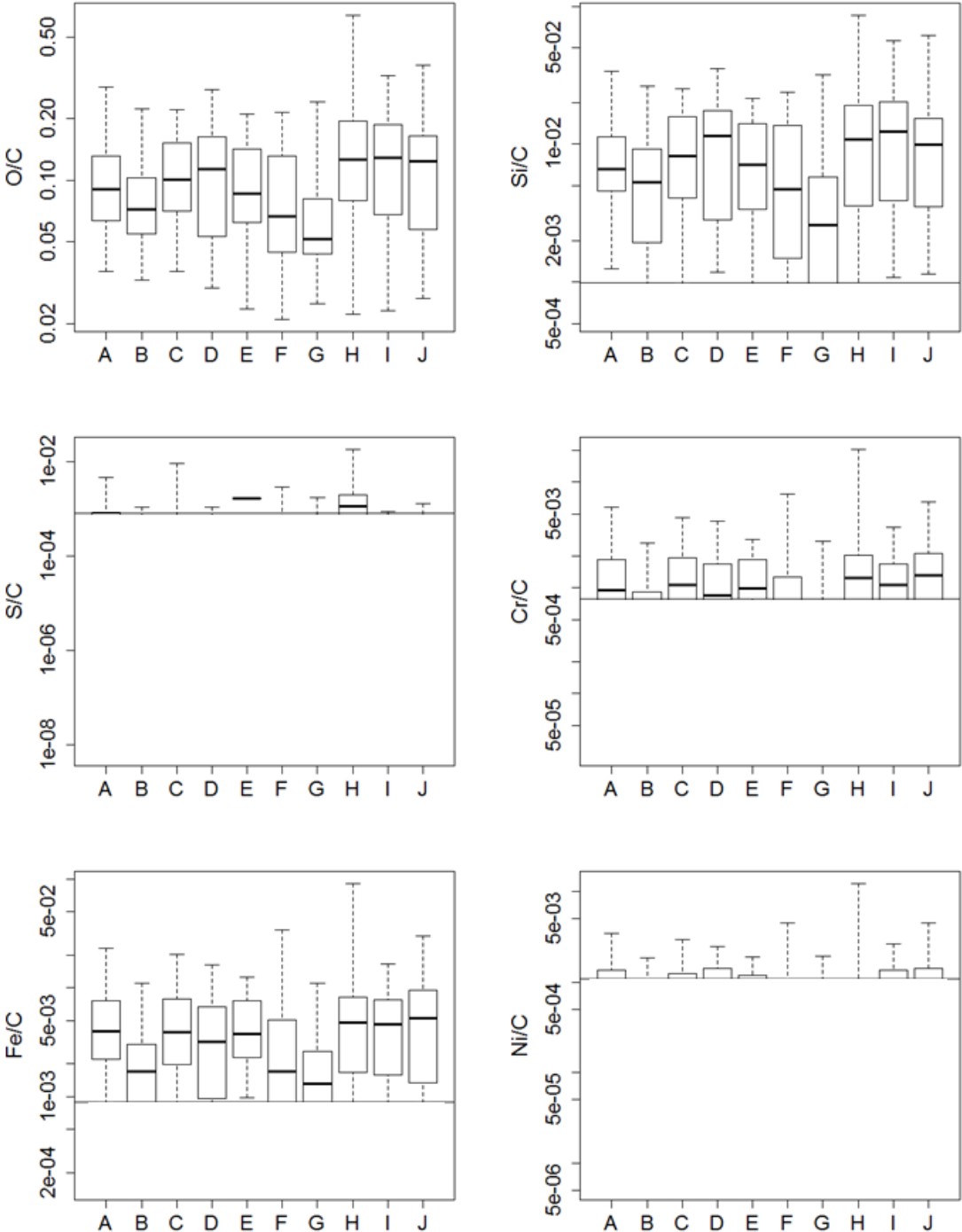

**Figure 6:** Censored boxplots of element ratios relative to C (atom %) determined by SEM-EDX (30 particles per sample). Sample K is not shown due to the different substrate used. Lower and upper quartiles appear as a box, minimum and maximum values as whiskers. Values below detection limit (horizontal line) are not shown.

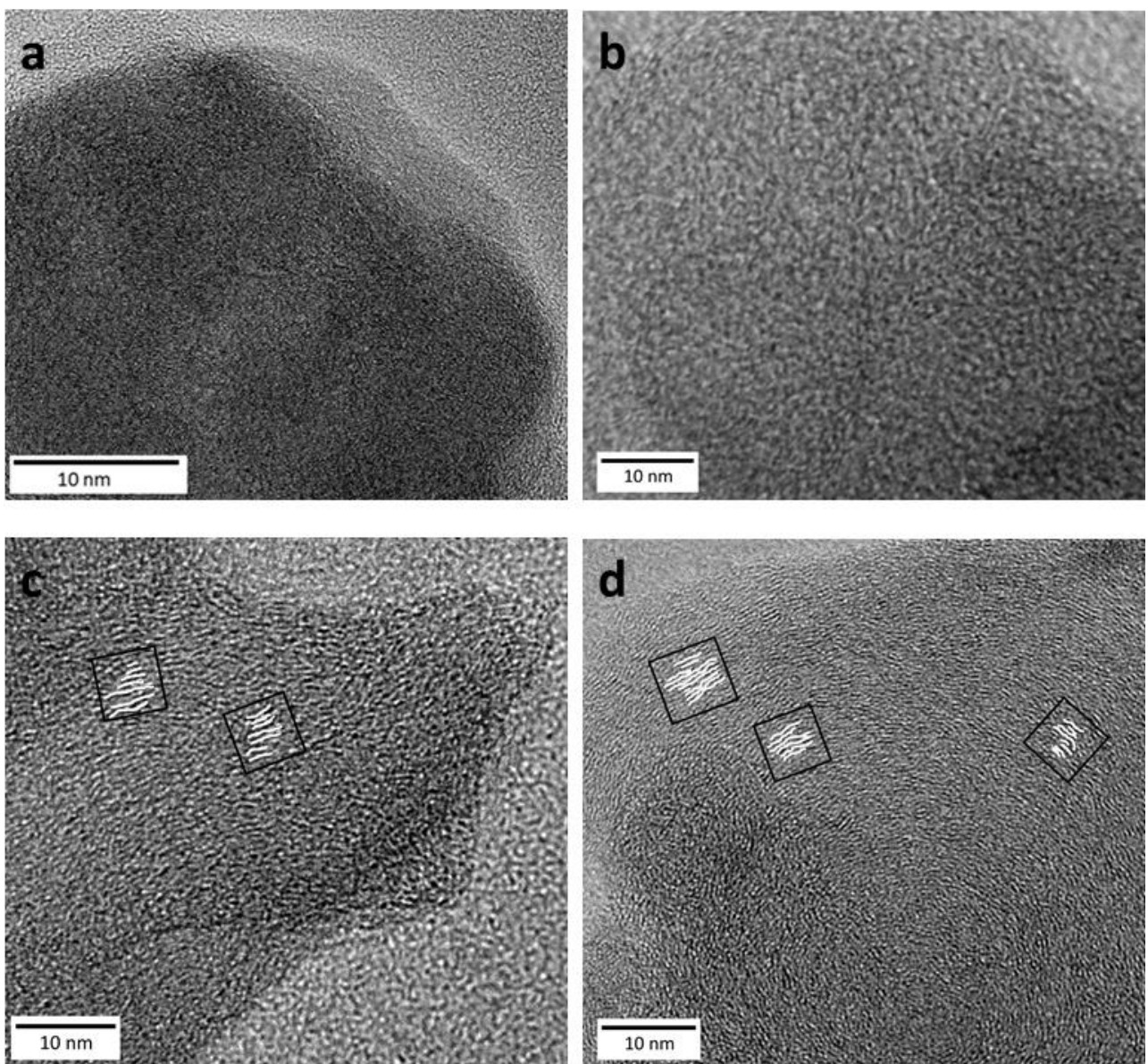

**Figure 7**: High resolution TEM image of individual refractory carbonaceous particles from sample G (17.4 km altitude): (a, b) completely amorphous particles, (c, d) particles showing small regions with ordering.

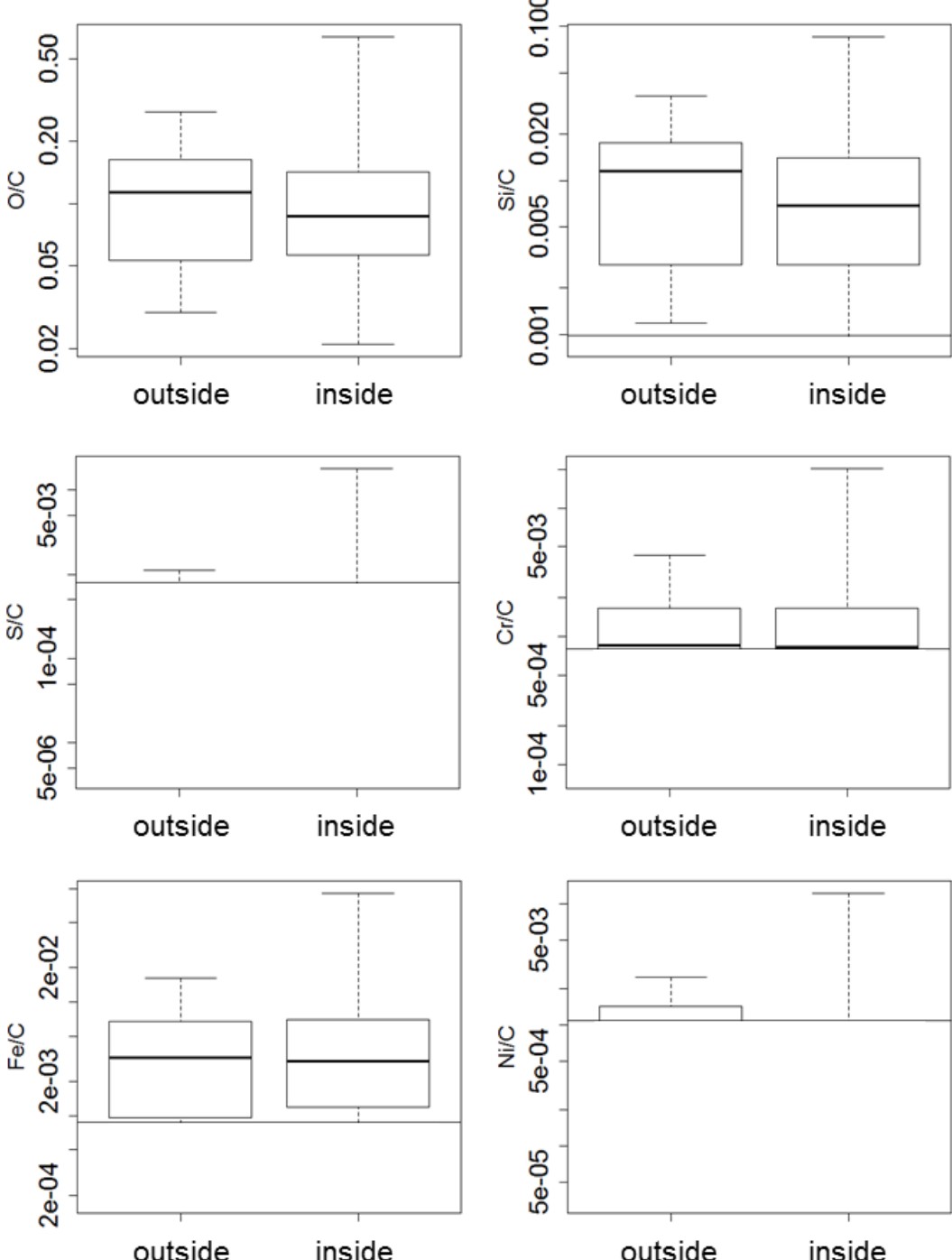

**Figure 8:** Censored boxplots of element ratios (atom %) relative to C, separately for outside and inside the polar vortex. Sample K was excluded from the analysis due to the different substrate used. Lower and upper quartiles appear as a box, minimum and maximum values as whiskers. Values below detection limit (horizontal line) are not shown.