# Peer review of "Sub-micrometer refractory carbonaceous particles in the polar stratosphere"

_Atmospheric Chemistry and Physics, 2017_

## Referee Comment (RC1) · Anonymous Referee #1 · 1 May 2017

This manuscript describes electron microscopy of particles sampled from the polar stratosphere. It posits the existence of a new class of particles there, small refractory particles with high carbon content. If correct, these measurements are important for understanding aerosols in the polar stratosphere. For reasons given below, the data are quite implausible, but neither is there anything definitely wrong. I have comments in three areas: technique, plausibility, and minor comments.

If this is published, it should be more in the tone of "we have some observations we don't trust and definitely don't understand, but they are the only observations we have".

1) Technique.

[Figure]

a) The samples are from 2000, and the analyses appear recent. Does this mean the samples were stored in plastic boxes at room temperature for roughly 15 years before analysis? It might not be a problem, or there could possibly be artifacts from odd things like chemical interactions with vapors from the plastic box. Please comment in the manuscript.

b) Are there any control samples? The manuscript mentions filter blanks, which are different than controls. Confidence in the results would be higher if samples from a known environment (for example, in the troposphere) gave the known results.

c) Can you define "refractory" more quantitatively? If a particle doesn't evaporate under the electron beam in vacuum, what does that mean in terms of carbonaceous material? Something with just a moderately high molecular weight, or does "refractory" mean it is practically a carbonate rock? Where would typical secondary organic material fall?

2) Plausibility.

a) I don't understand the lack of sulfur in the particles. In Figure 7, for most particles sulfur is below the detection limit of about 0.1%. The polar stratosphere vortex is a region of condensation of $H_2SO_4$ that had evaporated and photolyzed in the upper stratosphere (Mills, 2005 and references therein). At the concentrations modeled by Mills et al., it would take perhaps a week for enough $H_2SO_4$ to condense on a 100 nm particle to produce more sulfur than observed. This is a rough calculation, and the calculations by Mills span quite a range. Still, it underscores the difficulty of explaining why sulfur is below detection limit on most of the particles. Any particle that spent a significant amount of time in the polar stratosphere should have some sulfate, and sub-100 nm particles with high surface-to-volume ratios should have measurable amounts of condensing material.

b) I also don't understand the lack of any difference in composition between the inside and outside of the vortex (abstract and Figure 9). Curtius et al. (2005) found some big differences in refractory fraction between the inside and outside of the vortex. Yet in this

manuscript there is no difference in composition. Nor is there any obvious correlation between the potential vorticity (PV) of samples with the volatile fraction (Table 1 and Figure 3). The samples listed with the highest volatile fraction (G and H) are inside the vortex, in apparent contradiction to Curtius et al.

c) Neither rocket exhaust nor extraterrestrial material are very consistent with the data. Kerosene rockets should produce soot, which was not observed. It is also doubtful that there are enough particulate emissions from non-solid-fuel rocket exhaust in the upper stratosphere to account for the observed particles. Note that the authors measured more carbonaceous material than meteor smoke. The authors could compare the estimated particulate emissions from rockets to the extraterrestrial flux to see if it is plausible that there is more carbonaceous material from rockets than meteoric smoke in the polar stratosphere. I tried a quick estimate and I don't think there is enough exhaust compared to meteor smoke, but I didn't spend much time tracking down references.

Considering extraterrestrial material, there is too little Mg, Si, and Fe compared to carbon. For example, the carbon to silicon atomic ratio in micrometeorites collected in the stratosphere is about 1.3 to 2.4, (Shramm et al., 1989) but in these data it is over 30 (Figure 9). Even "carbonaceous" meteorites have more Mg, Si, and Fe than observed in these particles. In addition, the carbon in a meteoroid that got hot would oxidize to $CO_2$ and not produce particles. Submicron meteoroids don't get all that hot. So the particles would have to come from a cloud of submicron, organic particles in space around the Earth. I'm not a space scientist, but it seems that if such a cloud existed it would be known from effects on the near-Earth space environment.

3) Minor comments.

a) The units are incorrect on the extraterrestrial mass flux (line 25 page 11).

b) The discussion of sources only includes primary sources. What about secondary organics? Depending on the answer to my question about refractory organics, I would

suggest that the authors consider secondary organic material more seriously, either formed in the troposphere or stratosphere.

c) Can the authors estimate a mass mixing ratio for these particles, even to an order of magnitude?

d) Comparisons to other data could benefit from more attention to the size ranges and definition of "refractory." For example, CN measurements consider a particle with 8 nm worth of nonvolatile material as refractory, these data would not. Both the Murphy et al. and Renard et al. data are only for larger particles than the ~100 nm particles described in this manuscript. The Zolensky et al. (1989) paper is about particles that are so much bigger it is just confusing to mention them.

e) The data could benefit from plotting the samples against a tracer or depth into the vortex instead of just "inside" and "outside", or color-coding them on existing plots. I believe that N2O was measured during SOLVE.

Curtius et al., Observations of meteoric material and implications for aerosol nucleation in the winter Arctic lower stratosphere derived from in situ particle measurements, ACP, 2005.

Mills et al., Photolysis of sulfuric acid vapor by visible light as a source of the polar stratospheric CN layer, JGR, 2005.

---

## Referee Comment (RC2) · Anonymous Referee #2 · 5 May 2017

**Review of the ACP manuscript acp-2017-278**
"Sub 500 nm refractory carbonaceous particles in the polar stratosphere"
by K. Schütze et al., 2017

The above manuscript deals with electron microscopy (TEM, SEM) analysis of stratospheric particles sampled mainly in the Arctic polar vortex. As such measurements are rare, the presentation of the measured data is well suited for ACP, even if the results are not totally conclusive. However, there are some points, which should be improved before publication, some work, but feasible.

General remarks:
The first thing, which immediately leaped out at me when reading the abstract, was the big difference between the time of sampling and the time of publication. The samples were taken in 2000, now we have 2017. When was the analysis done? If it was in recent years, how where the samples stored in-between? How might the particles have changed during this long storage time? If the analysis was performed shortly after sampling, why did the publication take so long? The authors have to address this issue in a new paragraph.

Secondly, concerning the samples, section 2.2. There are 11 samples, OK, but I got confused how many particles where analyzed with which method. Were some particles analyzed with both methods? Moreover, on page 4, line 21 you even mention STEM, which is not mentioned somewhere else. Was this an additional method? Then it should be listed in the abstract as well. To make it easier for the reader to understand what you did and not to put too much workload on you, I suggest to include another table, where the reader gets an overview how many particles where analyzed with which method (and detector).

Another point, there are many statements in the manuscript, which are not specific enough. This occurs quite often, when citing the literature (which might not be the fault of the present authors, maybe the original authors were not specific enough). I tried to list some examples of that below. Please have a look throughout the manuscript and improve the text.

Finally, there are another two important issues (measurement artifacts and particle aging), which are explained in the following section in detail.

Specific remarks:
Abstract:
- p. 1, l.15: "… approximately 28-82% of the particles are refractory carbonaceous ..."
This statement is not very specific. You can nearly find all fractions of refractory carbonaceous particles, well OK, everything is possible, not very useful, but how likely is that? Moreover, isn´t the range much smaller, 52-82% (Fig. 3), considering that sample G seems to be a special one?

- p. 1 l. 17: "20-830 nm" this contradicts the manuscript title, i.e. the "500 nm"

 - p. 1 l. 21: The ratios to C: It would help the reads imagination if you would provide the ratios as fraction, i.e. instead of for instance "0.001" use "1/1000". Same for the detection limits in section 2.2. As the first place is the abstract, showing your major findings, you should

also asses the meaning of these numbers, are they common or rather rare, what do they indicate, etc.

- p. 2 1. 25: Ebert et al. 2016, from the same group, what are the similarities, what are the differences between this paper and the current manuscript? It should be possible to compare the results.

- p. 3 1. 4: redistribution vs. sedimentation: Currently your statement reads like an exclusive "or", but both processes can happen to a specific trace gas, it can be redistributed and be removed by sedimentation, or?

- p. 3 1. 6: In this paragraph, you list a bunch of sources for stratospheric refractory particles. However, the reader does not know, which one is the more important (e.g. with respect to mass or frequency of occurrence). Could you please provide the reader with such an additional information.

- p. 4 1. 2: It is stated that the vortex was stable between mid-January and mid-March. This was exactly the time of sampling and you should mention here that the presented data are from this period.

- p. 4 1. 11: the selection criteria "substrate area covered by particles", what does it mean? I´m not an electron microscopy specialist.

- p. 4 sect. 2.2: The two silicon-EDX detectors from Oxford, are they the same? Once Oxfordshire, once Wiesbaden? Isn´t it the same company?

- p. 5 1. 10: Not being an electron microscopy specialist: what would you expect the scattered electrons do? Hit the housing and generate x-ray emissions there? Please clarify.

- p. 5 1. 17: "small but systematic differences". Please specify what "small" means, e.g. give a percentage range. Same for line 21.

- p. 6 1. 15: "all particles" in the world? The stratosphere? On a sample? Which diameter does the spot have?

- p. 6 1. 25: For me, sample G seems to be special. Did you check how the sampling conditions of sample G are compared to the other samples?

- p. 7 1. 1: In Fig 4, sample E and F show a very different distribution width. Did you check for reasons?

- p. 7 1. 7: The minor components you have found (Fig. 5): From the literature (e.g., Murphy et al., AS&T, 2004; Martinsson et al., AMT, 2014) it is well known that ice crystals hitting the aerosol inlet can remove inlet material, bring it into the air and thus can generate artificial particle signals. Fe and Ni are known for this. Did you check the correlation between the occurrence of these elements with the ice crystal number concentration or the sampling time spent in ice clouds? This is **important** and must be addressed in order to trust your data.

- p. 7 1. 24: For how many particles was this element distribution imagine done? Fig. 6 shows just one. Is there any statistics on the results of this analysis?

- p. 7 1. 25: I do not know how the element distribution images work, hence I do not know what "measuring several" (how many?) "points on the particles" means. Please explain this more in detail.

- p. 7 1. 31: Why did you generate these four groups? If I did not overlook it, they are not used afterwards.

- p. 9 1. 22: The differences between your study and the results in Nguyen et al., 2008 is likely due to the different atmospheric measurement regions and different measurement altitudes. You should mention that, otherwise the reader might take the Nguyen reference as a contradiction to your findings, which is, in my point of view, not the case.

- p. 10 1. 11: Pyro-convection is defined as fire-started or fire-added convection, hence the definition given by you is incomplete.

- p. 11 aircraft exhaust section: The Mazaheri et al. reference here, and later on also the Tumolva et al. and Torvela et al. references in the wood burning section, here you compare freshly emitted particle properties to your particles, which are, because they were measured in the polar vortex, likely more than one year old. This comparison can only fail, the particles aged and strongly changed. I miss this time effect in all potential source paragraphs, but this point is **important** and must be considered in the discussion section.

- p. 11 1. 26: Consider to add "(dominant meteorite fraction)" or something similar after "chondrites", in order to explain what this thing is.

- p. 15 1. 1: The summary is too short. You did a lot of work, please expand the summary.

Technical corrections:

- p. 1 1. 29: Please remove the empty line, the last sentence of the abstract belongs to the upstream paragraph and should not be separated.

- p. 2 l. 3: "sulfur" is an "element", not a "component".

- p. 2 1. 13: which "groups" where identified? "Particle" or "morphology" or …

- p. 2 1. 15: "a large refractory particle load", what does this mean? With respect to particle mass or particle number or just fraction of particles containing refractory material?

- p. 2 1. 20: "widely distributed", what does this mean? All over the globe? Or at all altitudes (which ones?) in the area of investigation (which was?)?

- p. 2 1. 22: I´m not a native speaker, but shouldn´t it be "Earth´s"?

- p. 2 1. 31: "condensation of saturated gases", it is not necessary to provide seven (!) references for this textbook process. As it disturbs reading the paper, you should reduce the number.

- p. 3 1. 25: Please insert a comma after "impactor".

- p. 3 1. 26: Please remove the "The" before "MACS".

- p. 3 1. 32: "It was weaker …" What is "it"? The "Arctic winter"? But then the sentence does not make sense.

- p. 4 1. 4: Please use "Θ" instead of "PT".

- p. 4 1. 20: Please move "software" before the brackets.

- p. 4 sect. 2.2: Please use "EDX" instead of "energy-dispersive X-ray" throughoutly, after you defined it once.

- p. 6 1. 29: The whole statistical analysis section reads like a bullet point list. Please make it more a coherent text or a real bullet point list, with an introductory text.

- p. 6 1. 7: Please remove "applying a significance level of 5%", this is redundant, as it is mentioned in the last sentence of this paragraph.

- p. 6 1. 15: Please move the comma after "(Figure 1)".

- p. 7 1. 1: Please use "indicated" instead of "shown", you do not show real particle size distributions, e.g. dN/dlogDp.

- p. 7 1. 6: Please move "besides C" to the beginning of the sentence.

- p. 7 1. 25: Please replace "contained in the whole" with "found everywhere in".

- p. 8 1. 18: Please replace "The samples" with "All samples".

- p. 10 1. 27: Please replace "emissions" with "eruptions".

- p. 11 1. 15: A space is missing before "The".

- p. 11 1. 16: Please replace "at" with "in".

- p. 12 1. 27: "comprised … to" sounds strange, better use "contribute …to" or something similar.

- Fig. 2: Please specify Kα and Kβ in the figure caption. What does "all particles" mean? In the stratosphere or all sample or all refractory? Is the peak height/area linearly representative for the number of atoms? This should be mentioned somewhere in the text.

- Fig. 3: The given particle numbers are the total number of analyzed particles or only the refractory ones? Please specify this "n" in the figure caption.

- Fig. 6: The colors in the lower row of pictures are hard to see. I believe to use bright red or even white as occurrence indicator color in all pictures would improve the figure.

---

## Short Comment (SC1) · 29 May 2017

**Comment on Schütze et al. (Atmos. Chem. Phys. Discuss., doi:10.5194/acp-2017-278, 2017)**

Alexander D. James (cmaj@leeds.ac.uk)

This article has the potential to be a useful addition to the understanding of stratospheric aerosol, particularly since there are relatively few capture and return samples with statistics on this level. Whilst the conclusions they are able to draw are limited, publication of such data is vital in facilitating future understanding. I feel that the authors have missed or omitted a section of the literature which, once considered, can both add to the understanding of the results and increase the potential audience of the article. Below are a number of specific comments on language, formatting and content.

The authors provide a good introduction to the current and historical field. I was surprised to see no reference to the recent and thorough review of Kremser et al. (2016), who summarised some of the studies mentioned in the introduction of this work and other related topics.

Page 4 line 2; change "extend" to read "extent".

The analysis of images for structure of carbonaceous material is interesting. Was electron diffraction data recorded for any samples?

Page 5 line 8; reformat $5 \times 10^{-3}$.

Page 5 line 27; amend to "too close to"

Page 5 line 30 onwards; sentence is hard to understand. Perhaps "Any particle which showed no signs of destruction or morphological change was taken to be non-volatile. Particles which changed under the electron beam were deemed volatile, allowing quantification of the fraction of aerosol which is volatile."

Section 4.1; I believe this section would benefit from also discussing the size ranges of the various particles. For example Ebert et al. (2016) discuss mainly particles of radius greater than 500 nm, which have metallic or meteoritic composition. In that study the smaller size fraction is described as being largely carbonaceous material in sulfate liquid droplets, similar to the findings of this study.

Page 12 line 1; change to "the particles described above matches the refractory…"

Possibility that particles have an extraterrestrial origin; this section makes a good comparison between measurements of extraterrestrial material and the particles observed in the stratosphere. What is lacking is any discussion of the process which occur as a result of frictional heating during atmospheric entry. There is currently some discussion of whether unablated meteoric material will sediment too rapidly to be found in the stratosphere (Carrillo-Sánchez et al., 2016), or whether significant fragmentation of ablating meteorites could lead to smaller aerosol with longer atmospheric lifetimes (Subasinghe et al., 2016). Considering these processes in the

light of the results presented here would broaden the appeal of the current results to a wider audience and add significantly to the conclusions the authors are able to draw from their data.

Regarding ablation: The fact that the three metals discussed; Ni, Fe and Cr; have ratios significantly different than their chondritic abundances has rather more interesting implications when considered with respect to the ablation process. Since in interplanetary dust Ni is largely contained in relatively volatile metal phases (melting points around 1200-1500 K), Fe is spread between volatile metallic and more refractory silicate phases (melting point >1800 K) and Cr is contained in the less volatile silicates (Bunch and Olsen, 1975), the three elements will ablate rather differently (Gómez-Martín et al., 2017). The relative volatility of Ni is therefore reconcilable with the larger Ni/Fe ratio measured here and does not rule out an extraterrestrial source, but the high Cr/Fe and Cr/Ni suggest that Cr at least has a terrestrial source, since if anything Cr should ablate less completely than the other elements.

Regarding fragmentation: This is hypothesised to happen by evaporation of volatile phases such as iron sulfides and amorphous carbonaceous material (ordered graphitic material would be much more refractory). It may be reasonable as a result that the metal bearing silicates would remain in larger particles which have very short lifetimes in the stratosphere, but carbonaceous material and some additional Fe is atmospherically available as a result.

The question could possibly be more constructively phrased in another way. Since we know that meteoric ablation occurs and meteoric smoke forms, why is it not unequivocally observed in these measurements? Indeed numerical modelling of MSPs suggests that they should be observable in this size range (Bardeen et al., 2008). Could it be that nucleation, growth and sedimentation of crystalline PSC has removed meteoric material? What implications would the partial dissolution of metals have on these measurements? Could dissolution, precipitation and agglomeration in liquid droplets cause more rapid growth of MSP compared to model predictions?

These comments also inherently include the issue of sample aging, which both anonymous reviewers rightly mention. Having some experience of electron microscopy, I suspect that this statistical detail could only be reached from measurements which would take several years to make. In addition to the reviewer's comments then, the possibility should be considered that some samples have aged more than others.

References

Bardeen, C. G., Toon, O. B., Jensen, E. J., Marsh, D. R., and Harvey, V. L.: Numerical simulations of the three-dimensional distribution of meteoric dust in the mesosphere and upper stratosphere, J. Geophys. Res.: Atmos., 113, D17202, 2008.

Bunch, T. E., and Olsen, E.: Distribution and significance of chromium in meteorites, Geochim. Cosmochim. Acta, 39, 911-927, http://dx.doi.org/10.1016/0016-7037(75)90037-X, 1975.

Carrillo-Sánchez, J. D., Nesvorný, D., Pokorný, P., Janches, D., and Plane, J. M. C.: Sources of cosmic dust in the Earth's atmosphere, Geophys. Res. Lett., 43, 11,979-911,986, 10.1002/2016GL071697, 2016.

Ebert, M., Weigel, R., Kandler, K., Günther, G., Molleker, S., Grooß, J. U., Vogel, B., Weinbruch, S., and Borrmann, S.: Chemical analysis of refractory stratospheric aerosol particles collected within the arctic vortex and inside polar stratospheric clouds, Atmos. Chem. Phys., 16, 8405-8421, 2016.

Gómez-Martín, J. C., Bones, D. L., Carrillo-Sánchez, J. D., James, A. D., Trigo-Rodríguez, J. M., B. Fegley, J., and Plane, J. M. C.: Novel experimental simulations of the atmospheric injection of meteoric metals, Astrophys. J., 836, 212, 2017.

Kremser, S., Thomason, L. W., von Hobe, M., Hermann, M., Deshler, T., Timmreck, C., Toohey, M., Stenke, A., Schwarz, J. P., Weigel, R., Fueglistaler, S., Prata, F. J., Vernier, J. P., Schlager, H., Barnes, J. E., Antuña-Marrero, J.-C., Fairlie, D., Palm, M., Mahieu, E., Notholt, J., Rex, M., Bingen, C., Vanhellemont, F., Bourassa, A., Plane, J. M. C., Klocke, D., Carn, S. A., Clarisse, L., Trickl, T., Neely, R., James, A. D., Rieger, L., Wilson, J. C., and Meland, B.: Stratospheric aerosol—Observations, processes, and impact on climate, Rev. Geophys., 54, 278-335, 2016.

Subasinghe, D., Campbell-Brown, M. D., and Stokan, E.: Physical characteristics of faint meteors by light curve and high-resolution observations, and the implications for parent bodies, Mon. Not. Royal Astro. Soc., 457, 1289-1298, 2016.

---

## Author Comment (AC1) · 30 Aug 2017

We gratefully acknowledge the suggestions of the anonymous Referee I and included them to the revised version of the paper. We believe that the changes considerably helped to improve the quality of the manuscript.

Anonymous Referee #1

This manuscript describes electron microscopy of particles sampled from the polar stratosphere. It posits the existence of a new class of particles there, small refractory particles with high carbon content. If correct, these measurements are important for understanding aerosols in the polar stratosphere. For reasons given below, the data are quite implausible, but neither is there anything definitely wrong. I have comments in

three areas: technique, plausibility, and minor comments. If this is published, it should be more in the tone of "we have some observations we don't trust and definitely don't understand, but they are the only observations we have".

1) Technique.

a) The samples are from 2000, and the analyses appear recent. Does this mean the samples were stored in plastic boxes at room temperature for roughly 15 years before analysis? It might not be a problem, or there could possibly be artifacts from odd things like chemical interactions with vapors from the plastic box. Please comment in the manuscript.

The samples were collected in 2000, analysis started late in 2013. We found the samples after storage in a desiccator and found them worth to be analyzed since data on stratospheric particles are sparse. As we also investigated blank samples, which were packed in the same sampling device (MACS), and stored in the same way as the real samples, we can exclude contamination from vapors of the plastic box or other possible artifacts related to storage. This issue is accordingly addressed in the last paragraph of chapter 2.1 in the paper: "The stratospheric particle samples (deposited on TEM grids) taken within the polar vortex, were packed into single plastic boxes and stored in a desiccator prior to analysis, starting in 2014. Based on the investigation of blank samples, contamination of the samples during the time of storage (e.g. by vapours from the plastic boxes) can be excluded. Furthermore, a change in particle morphology and nanostructure is not expected, since the particles found are either amorphous or show very little ordering. This conclusion is based on the fact that graphitization of carbonaceous material is an irreversible process (Diessel et al., 1978; Itaya, 1981; Pesquera and Velasco, 1988). Anyhow, it should be kept in mind that other parameters (chemical composition, mixing state) may be changed to a variable extent by aging."

b) Are there any control samples? The manuscript mentions filter blanks, which are different than controls. Confidence in the results would be higher if samples from a

known environment (for example, in the troposphere) gave the known results.

Unfortunately there are no control samples available. This is something we are aware of, and changed in recent campaigns, but can't change for past actions. The filter blanks the reviewer mentions proceeded the same actions as the real samples, e.g. equipping to the MACS, taking part in the sampling procedure without being exposed to ambient air, disassembly, storage, handling for measurements and measuring the samples themselves.

c) Can you define "refractory" more quantitatively? If a particle doesn't evaporate under the electron beam in vacuum, what does that mean in terms of carbonaceous material? Something with just a moderately high molecular weight, or does "refractory" mean it is practically a carbonate rock? Where would typical secondary organic material fall?

We do know many carbonaceous materials, e.g. soot, spores, fragments of plants,... which are stable under the high vacuum conditions in SEM and under the electron beam and which are thus identified to be refractory. But based on both the size and morphology of the particles, we can exclude all those particle species to be the same as the particles found in the current study. Furthermore, we do know that highly volatile material evaporates under both conditions in SEM mentioned. But unfortunately it is difficult to quantify the real nature of the carbonaceous particles. Regarding the definition of "refractory" we added a sentence to chapter 2.2: ". Similar to Ebert et al., 2016, we have classified all particles that are stable (no visible morphological change) under the high vacuum conditions and electron beam excitation in the SEM and TEM as refractory." Based on our long-lasting experience with tropospheric particles we know, that secondary organic materials (except soot) easily evaporates under the electron beam and does, thus, not show any "refractory" behavior. In addition, it would be questionable where the minor amounts of the elements Fe/Cr/Ni/Si found in the current study should come from in case of condensation (?) of secondary organic material.

2) Plausibility.

a) I don't understand the lack of sulfur in the particles. In Figure 7, for most particles sulfur is below the detection limit of about 0.1%. The polar stratosphere vortex is a region of condensation of H2SO4 that had evaporated and photolyzed in the upper stratosphere (Mills, 2005 and references therein). At the concentrations modeled by Mills et al., it would take perhaps a week for enough H2SO4 to condense on a 100 nm particle to produce more sulfur than observed. This is a rough calculation, and the calculations by Mills span quite a range. Still, it underscores the difficulty of explaining why sulfur is below detection limit on most of the particles. Any particle that spent a significant amount of time in the polar stratosphere should have some sulfate, and sub-100 nm particles with high surface-to-volume ratios should have measurable amounts of condensing material.

We agree. As counting statistics might bear large errors based on impaction of several liquid H2SO4-droplets on the same spot, fast evaporation of small particles as well as fast evaporation in the vacuum chamber in the electron microscope leading to an underestimation of volatile particles, we decided to remove (the old) figure 3 from the manuscript. Furthermore, also very thin layers of volatile material on some of the re-fractory particles may have vaporized under the high vacuum conditions in the electron microscope. As each sample was at least equipped to the microscope for three times (first scan, measurement by SEM, measurement by TEM), this effect is likely to occur and may explain the low amount of S in (the new) figure 6. Anyhow, we are aware of the fact, that many more of the externally mixed carbonaceous particles should show some sulfur coating and thus discuss this issue in the text: "Most of the refractory carbonaceous particles are not included in or coated by sulfate. This is surprising, as the particles were sampled in air having low abundance of N2O and therefore long residence times in the stratosphere (Table 1). Therefore, one would expect that all refractory particles occurring in the polar stratosphere are covered by sulfuric or nitric acid. The low abundance of refractory particles internally mixed with sulfates contra-dicts expectations based on the models by Mills et al. (2005) as well as the findings of Sheridan et al. (1994) and Murphy et al. (2013) which suggest that most or all stratospheric refractory particles should be embedded in or coated with sulfuric acid. The results of our study can partly be explained by the evaporation of the sulfate component in the electron beam prior to its identification. The mixing state of the refractory carbonaceous particles may also be caused by splattering of volatile material of previously internally mixed refractory/volatile material. However, the reason for most of the refractory carbonaceous particles to be externally mixed remains open."

b) I also don't understand the lack of any difference in composition between the inside and outside of the vortex (abstract and Figure 9). Curtius et al. (2005) found some big differences in refractory fraction between the inside and outside of the vortex. Yet in this manuscript there is no difference in composition. Nor is there any obvious correlation between the potential vorticity (PV) of samples with the volatile fraction (Table 1 and Figure 3). The samples listed with the highest volatile fraction (G and H) are inside the vortex, in apparent contradiction to Curtius et al.

The comparison of our data deduced from SEM-measurements are probably difficult to compare to COPAS-data from Curtius et al. (2005). The Aerosol preheater operated with COPAS forces volatile aerosol compounds to evaporate within 1 -2 seconds while exposed to temperatures of $\sim250°C$ – that this technique works to evaporate H2SO4 of stratospheric sizes to sizes below the COPAS detection limit was demonstrated to perform well (cf. Weigel et al. 2009). This kind of "flash"-evaporation by using thermo-denuders may force any solvent within a H2SO4 particle to get stuck together (including also insoluble incorporations) due to the surface tension of the evaporating H2SO4. The identical instrument as used by Curtius et al 2005 was also deployed for the refining study Weigel et al. 2014. Thus a comparison of results seem to be most appropriate between these studies only.

The impactor sample technique and offline analysis, in contrast, may imply a very different process of separating volatile (mostly liquid) substances from incorporations of the stratospheric aerosol. The mostly liquid stratospheric particle literally smashes on the impaction plate such that liquid compounds with solvents as well as solid incorporations are dispersed over a certain impact area on the substrate. Thus, the contributions of H2SO4 solvents may remain included in the H2SO4 matrix while all insoluble incorporation of the initial stratospheric aerosol particle are present as kind of debris field on the substrate surface to be analyzed.

In essence, the contribution (by number) of solvents to the "flash"-evaporized residual (COPAS technique) is not quantifiable and may vary from measurements in- and outside of the polar vortex. Such differences may be detectable when measured with 1Hz resolution (COPAS). Such differences (by number) of refractory aerosol, however, may not be resolved by aerosol samples in- and outside the polar vortex (due to e.g. sampling times over minutes) which was, furthermore, not the intention to investigate with this study, but instead, to investigate potential differences in the chemical composition and/or physical nature of particles sampled in- or outside the polar vortex.

As explained in 2a) we removed figure 3 from the manuscript.

c) Neither rocket exhaust nor extraterrestrial material are very consistent with the data. Kerosene rockets should produce soot, which was not observed. It is also doubtful that there are enough particulate emissions from non-solid-fuel rocket exhaust in the upper stratosphere to account for the observed particles. Note that the authors measured more carbonaceous material than meteor smoke. The authors could compare the estimated particulate emissions from rockets to the extraterrestrial flux to see if it is plausible that there is more carbonaceous material from rockets than meteoric smoke in the polar stratosphere. I tried a quick estimate and I don't think there is enough exhaust compared to meteor smoke, but I didn't spend much time tracking down references. Considering extraterrestrial material, there is too little Mg, Si, and Fe compared to carbon. For example, the carbon to silicon atomic ratio in micrometeorites collected in the stratosphere is about 1.3 to 2.4, (Shramm et al., 1989) but in these data it is over 30 (Figure 9). Even "carbonaceous" meteorites have more Mg, Si, and Fe than observed in these particles. In addition, the carbon in a meteoroid that got hot would oxidize to $CO_2$ and not produce particles. Submicron meteoroids don't get all that hot.

So the particles would have to come from a cloud of submicron, organic particles in space around the Earth. I'm not a space scientist, but it seems that if such a cloud existed it would be known from effects on the near-Earth space environment.

Regarding rocket exhaust we have changed the last sentences to: "As the refractory carbonaceous particles observed by us are not soot, their origin from rocket exhaust is unlikely. However, as carbonaceous rocket exhaust particles were not investigated previously by electron microscopy this source cannot be excluded." Based on the interactive comment posted by Alexander D. James, we added a new paragraph to the "extraterrestrial material" section: "The chemical composition of extraterrestrial material may be strongly fractionated by frictional heating during atmospheric entry (e.g., Carrillo-Sánchez et al., 2016; Gómez- Martin et al., 2017). The processes taking place during atmospheric entry include ablation by sputtering and thermal evaporation as well as fragmentation. Meteorite ablation particles usually occur as iron, glass or silicate spheres (e.g., Blanchard et al., 1980; Murrell et al., 1980). Submicrometer refractory carbonaceous particles resulting from meteoric ablation and fragmentation have - to the best of our knowledge - not been described in previous literature. However, it is conceivable that such particles originate from carbonaceous material contained in meteorites or interplanetary dust particles. The observed non-chondritic ratios of the minor elements Fe, Cr, Ni are not a strong argument against such an origin, as the volatility of these elements depends on the minerals in which they are contained. Most of extraterrestrial Fe occurs as metal, silicate or oxide, most of Ni as metal (Papike, 1998). Cr may occur as oxide, sulphide or nitride as well as a minor component in metal and silicates (Bunch and Olsen, 1975). Depending on the relative abundance of the different mineral phases, substantial fractionation during evaporation can be expected (see also Floss et al., 1996). In summary, meteoric ablation and fragmentation particles are a possible source of the particles encountered in the present study."

3) Minor comments.

a) The units are incorrect on the extraterrestrial mass flux (line 25 page 11).

[Figure]

The unit was corrected to 5-270 tons per day.

b) The discussion of sources only includes primary sources. What about secondary organics? Depending on the answer to my question about refractory organics, I would suggest that the authors consider secondary organic material more seriously, either formed in the troposphere or stratosphere.

We agree that secondary organic material, either formed in the troposphere or stratosphere, can serve as a source for carbonaceous particles in the stratosphere. As mentioned in 1c) the secondary material we know from our experience with tropospheric samples is highly volatile under high vacuum conditions and evaporates rapidly when measured by EDX. If the same processes and/or precursor gases make up secondary organic material in the stratosphere, then we expect the probability of the particles to originate from SOA to be low. As we cannot totally deny this possibility, we have changed p. 10 l. 19 based on Reviewer 2 to "Mixed carbon-sulfur particles were observed by Nguyen et al. (2008) (diameter $\leq 1$ $\mu$m) at 10 km altitude between 50°N and 30°S. These particles were assumed to have formed from condensed organic matter. The differences between these particles and those found in the current study might result from differences in sampling altitudes and regions. Therefore we cannot totally exclude the particles to be different, taking into account that the particles might have evolved from condensed organic matter. However, we do not know if secondary organic particles become refractory as a result of atmospheric processes."

c) Can the authors estimate a mass mixing ratio for these particles, even to an order of magnitude?

Based on the impactor flow, we can roughly estimate a mass mixing ratio of the particles. Again, we have to emphasize that the errors of this estimate are large. The mass mixing ratio of the refractory carbonaceous particles varies between 0.65 (sample B) and 2.3 (mg air)-1 (sample D) with a median for all samples of 1.1 (mg air)-1. We included one additional paragraph in the introductory part of chapter 3: "Given the size of

the refractory particles and the performance of the impactor, all similar particles in the sampled air were likely delivered to the impactor and collected there. Since the amount of air drawn through the impactor is known, the atmospheric abundance of these particles can be estimated from the number of particles in the impactor sample. That number was estimated from electron micrographs sampling the impaction spot and the size if the impaction spot. The ambient number mixing ratio of the refractory carbonaceous particles varies between 0.65 (mg air)-1 and 2.3 (mg air)-1, with a median for all samples of 1.1 (mg air)-1 (Table 1). When compared with CPC measurements in Table 1, the carbonaceous particles comprised a few percent of the total number of particles in the air." Furthermore we discuss the mass flux at the end of section 4.1: "In summary, the sole occurrence of refractory carbonaceous particles and sulfates in stratospheric samples was reported in previous literature but seems to be uncommon. The median number mixing ratio (1.1 mg air-1) of carbonaceous particles is smaller by an order of magnitude than the abundance of non-volatile particles reported by, e.g., Weigel et al. (2014) for measurements in the winter stratospheric polar vortex in 2003, 2010 and 2011. The method described by Weigel et al. involves exposure of particles to a temperature >250 °C and determination (with a CPC) of the number of particles that did not evaporate to sizes below the detection limit of the CPC. They concluded that up to 80% of the particles present were non-volatile by this criterion. Following our definition only a few percent of the SOLVE particles are non-volatile in the electron microscope. This discrepancy may be caused by the different definitions of a non-volatile particle."

d) Comparisons to other data could benefit from more attention to the size ranges and definition of "refractory." For example, CN measurements consider a particle with 8 nm worth of nonvolatile material as refractory, these data would not. Both the Murphy et al. and Renard et al. data are only for larger particles than the ∼100 nm particles described in this manuscript. The Zolensky et al. (1989) paper is about particles that are so much bigger it is just confusing to mention them.
We agree that the particles detected by Murphy et al. regard basically size ranges ≥200 nm, up to several micrometers and those described by Renard et al. (2008) up to 350 nm. In more recent papers (Murphy et al., 2013) the authors were able to detect particles from 120 nm. Therefore the size ranges by the authors mentioned were added in the introduction, as well as in the discussion. Anyhow, an overlap of the size ranges of particles exist for sizes between ∼120 − 800 nm and therefore we find the comparison of data by the mentioned authors meaningful. As the introduction is about general sources of particles in the stratosphere, we decided to leave the paragraph regarding rocket exhaust from Zolensky et al. (1989). In the paragraph regarding rocket exhaust as a potential source for stratospheric particles, the comparison to Zolensky et al. is used in the manner of showing, that rocket exhaust primarily emits characteristic $Al_2O_3$ spheres, which were not found in this study. There we do not compare them to carbonaceous particles but give a general statement on typical emissions by rockets. Therefore we also intended to leave the citation where it is.

e) The data could benefit from plotting the samples against a tracer or depth into the vortex instead of just "inside" and "outside", or color-coding them on existing plots. I believe that N2O was measured during SOLVE.

We agree. For clarity, we added the N2O-values in Table 1.

Curtius et al., Observations of meteoric material and implications for aerosol nucleation in the winter Arctic lower stratosphere derived from in situ particle measurements, ACP, 2005.

Mills et al., Photolysis of sulfuric acid vapor by visible light as a source of the polar tratospheric CN layer, JGR, 2005.

[Figure]

2017.

---

## Author Comment (AC2) · 30 Aug 2017

We gratefully acknowledge the suggestions of the anonymous Referee II and included them to the revised version of the paper. We believe that the changes considerably helped to improve the quality of the manuscript.

Anonymous Referee #2

The above manuscript deals with electron microscopy (TEM, SEM) analysis of stratospheric particles sampled mainly in the Arctic polar vortex. As such measurements are rare, the presentation of the measured data is well suited for ACP, even if the results are not totally conclusive. However, there are some points, which should be improved before publication, some work, but feasible.

[Figure]

General remarks:

The first thing, which immediately leaped out at me when reading the abstract, was the big difference between the time of sampling and the time of publication. The samples were taken in 2000, now we have 2017. When was the analysis done? If it was in recent years, how where the samples stored in-between? How might the particles have changed during this long storage time? If the analysis was performed shortly after sampling, why did the publication take so long? The authors have to address this issue in a new paragraph.

The samples were collected in 2000, analysis started late in 2013. The samples were stored in a desiccator and we found them worth to be analyzed since data on stratospheric particles are sparse. As we also investigated blank samples, which were packed in the same sampling device (MACS), and stored in the same way as the real samples, we can exclude contamination from vapors of the plastic box or other possible artifacts related to storage. This issue is accordingly addressed in the last paragraph of chapter 2.1 in the paper: "The stratospheric particle samples (deposited on TEM grids) taken within the polar vortex, were packed into single plastic boxes and stored in a desiccator prior to analysis, starting in 2014. Based on the investigation of blank samples, contamination of the samples during the time of storage (e.g. by vapours from the plastic boxes) can be excluded. Furthermore, a change in particle morphology and nanostructure is not expected, since the particles found are either amorphous or show very little ordering. This conclusion is based on the fact that graphitization of carbonaceous material is an irreversible process (Diessel et al., 1978; Itaya, 1981; Pesquera and Velasco, 1988). Anyhow, it should be kept in mind that other parameters (chemical composition, mixing state) may be changed to a variable extent by aging."

Secondly, concerning the samples, section 2.2. There are 11 samples, OK, but I got confused how many particles where analyzed with which method. Were some particles analyzed with both methods? Moreover, on page 4, line 21 you even mention STEM, which is not mentioned somewhere else. Was this an additional method? Then it

should be listed in the abstract as well. To make it easier for the reader to understand what you did and not to put too much workload on you, I suggest to include another table, where the reader gets an overview how many particles where analyzed with which method (and detector).

We agree that the reader gets confused about how many particles were analyzed with which method. Therefore we added a new table 2 at the beginning of chapter 2.2. STEM is an additional tool in TEM with the opportunity to get high-resolution information on the element distribution within nanoparticles. STEM images are shown in figure 6 and were mentioned in the discussion paper on p.7 l. 24. We also added the 4 particles investigated by STEM and the 23 particles where the nanostructure was investigated to the total number of investigated particles (4202) in the introduction as well as chapter 2.2.

Another point, there are many statements in the manuscript, which are not specific enough. This occurs quite often, when citing the literature (which might not be the fault of the present authors, maybe the original authors were not specific enough). I tried to list some examples of that below. Please have a look throughout the manuscript and improve the text. Finally, there are another two important issues (measurement artifacts and particle aging), which are explained in the following section in detail. Specific remarks: Abstract: - p. 1, l.15: "... approximately 28-82% of the particles are refractory carbonaceous ..." This statement is not very specific. You can nearly find all fractions of refractory carbonaceous particles, well OK, everything is possible, not very useful, but how likely is that? Moreover, isn't the range much smaller, 52-82% (Fig. 3), considering that sample G seems to be a special one?

After studying the reviews of both reviewers, we figured out that we are introducing large errors especially by stating relative numbers of refractory and volatile material. We probably highly underscore volatile particles because of a) losses during sampling (e.g. bounce off), b) losses under the high vacuum conditions under the electron microscope and c) errors in counting because of the impaction of more than one particle

on the same spot. Therefore we decided to remove figure 3 from the manuscript and thus the relative occurrence of refractory and volatile particles.

- p. 1 1. 17: "20-830 nm" this contradicts the manuscript title, i.e. the "500 nm

We agree and changed the title to "Sub-micrometer refractory carbonaceous particles in the polar stratosphere"

- p. 1 1. 21: The ratios to C: It would help the reads imagination if you would provide the ratios as fraction, i.e. instead of for instance "0.001" use "1/1000". Same for the detection limits in section 2.2. As the first place is the abstract, showing your major findings, you should also asses the meaning of these numbers, are they common or rather rare, what do they indicate, etc.

We do not agree and prefer "0.001" for clarity reasons. The numbers are given for the reader as a summary of the findings. We did this to give the reader as many information on the particles as possible, as we faced that in the current literature necessary information on particles is often missing in order to be able to compare data. Based on that fact, publications with information on element ratios are rare in literature. This also counts for descriptions of particles for specific sources. As this information is missing in literature, it is not possible to compare the data or give information what the numbers exactly indicate.

- p. 2 1. 25: Ebert et al. 2016, from the same group, what are the similarities, what are the differences between this paper and the current manuscript? It should be possible to compare the results.

The results from Ebert et al. 2016 are totally different to the ones from this study, as those authors found, besides the volatile particle group, eight different particle groups. The differences of Ebert et al. (and other authors) to the findings of the current study are discussed in the second last paragraph of section 4.1: "In the present study, only carbonaceous particles and sulfates were observed similar to previous findings

(Pueschel et al., 1992; Blake and Kato, 1995; Strawa et al., 1999; Nguyen et al., 2008). There are, however, several previous publications which describe the presence of a variety of other refractory particle groups in addition to carbonaceous particles. These additional particle groups include metallic particles (Chuan and Woods, 1984; Sheridan et al., 1994; Chen et al., 1998; Baumgardner et al., 2004; Ebert et al., 2016), meteoritic particles (Murphy et al., 1998, 2007, 2013; Renard et al., 2008, Ebert et al., 2016), silicates (Testa et al., 1990; Ebert et al., 2016), crustal-type particles (Sheridan et al., 1994; Chen et al., 1998), as well as Ca-bearing particles (Della Corte et al., 2013; Ebert et al., 2016)."

- p. 3 l. 4: redistribution vs. sedimentation: Currently your statement reads like an exclusive "or", but both processes can happen to a specific trace gas, it can be redistributed and be removed by sedimentation, or?

We agree, and changed the paragraph accordingly.

- p. 3 l. 6: In this paragraph, you list a bunch of sources for stratospheric refractory particles. However, the reader does not know, which one is the more important (e.g. with respect to mass or frequency of occurrence). Could you please provide the reader with such an additional information.

In the paragraph mentioned, we list the most probable sources for refractory stratospheric particles. Unfortunately no publications exist on the frequency of contribution from different sources to the refractory particle load. This is most probably the case because most of the sources do not occur continuously but are rather irregular features. Therefore we added an additional sentence to provide the reader with this information: "As the frequency of particle emissions from the listed sources is highly variable, the individual contribution of the various sources is, in general, not quantifiable"

- p. 4 l. 2: It is stated that the vortex was stable between mid-January and mid-March. This was exactly the time of sampling and you should mention here that the presented data are from this period.

Accordingly the sentence was changed to "During the period of airborne measurement operations, from mid-January on, the vortex evolved to be continuous and stable until mid-March (Greenblatt et al., 2002)."

- p. 4 1. 11: the selection criteria "substrate area covered by particles", what does it mean? I'm not an electron microscopy specialist.

For analysis by electron microscopy it is important to achieve a suitable particle load on the substrates. Particle overloading means too many particles in one certain area, which makes it impossible to identify and characterize individual particles. Thus, the mixing state cannot be doubtlessly identified and it is not possible to identify individual particles. Too little particles in a certain area of the impaction spot have too little material to achieve a representative number of analyzed particles.

- p. 4 sect. 2.2: The two silicon-EDX detectors from Oxford, are they the same? Once Oxfordshire, once Wiesbaden? Isn't it the same company?

Yes, both EDX-detectors are the same! Therefore we changed the sentence to: "The instrument is equipped with the same type of EDX-detector as the Philips CM20 instrument."

- p. 5 1. 10: Not being an electron microscopy specialist: what would you expect the scattered electrons do? Hit the housing and generate x-ray emissions there? Please clarify.

According to your statement we included the following sentence for clarity: "This could lead to the detection of chemical elements in the vacuum chamber's housing material."

- p. 5 1. 17: "small but systematic differences". Please specify what "small" means, e.g. give a percentage range. Same for line 21.

In order to show what the differences between the measurements with the differences are, we introduced a new table for the electronic supplement; S2. In this table the median values for element ratios are shown both for SEM and TEM.

- p. 6 1. 15: "all particles" in the world? The stratosphere? On a sample? Which diameter does the spot have?

The sentence was accordingly changed to: "All collected particles are located within a characteristic impaction spot having a diameter of $\sim$350 $\mu$m."

- p. 6 1. 25: For me, sample G seems to be special. Did you check how the sampling conditions of sample G are compared to the other samples?

We checked the conditions of sample G and also found an error in Table 1 which we changed (wrong order of PV-values). Thus we figured out, that sample G is from a multiple level flight and shows the lowest potential vorticity (18.2 PVU) and the second highest N2O value (209 ppbv). Furthermore sample G was one of the samples collected in the lowest altitudes (17.4 km) and shows, thus, a comparably high pressure and low potential temperature. Based on those facts we changed the values in table 1 and also added values for N2O.

- p. 7 1. 1: In Fig 4, sample E and F show a very different distribution width. Did you check for reasons?

We do not have any answer to this question since we do not have any idea what that difference in distribution width between the samples is due to.

- p. 7 1. 7: The minor components you have found (Fig. 5): From the literature (e.g., Murphy et al., AS&T, 2004; Martinsson et al., AMT, 2014) it is well known that ice crystals hitting the aerosol inlet can remove inlet material, bring it into the air and thus can generate artificial particle signals. Fe and Ni are known for this. Did you check the correlation between the occurrence of these elements with the ice crystal number concentration or the sampling time spent in ice clouds? This is important and must be addressed in order to trust your data.

The elements Fe, Cr and Ni are only abundant as minor elements in carbonaceous particles. Furthermore, STEM images reveal that those particles do not occur as inclusions in the particles but as small traces widely distributed within the particles. In contrast, particles both described by Murphy et al., 2004 and Martinsson et al., 2014 are solely consisting of all or some of the elements Fe, Cr or Ni. Thus we regard our particles as totally different as the ones found by those authors. Abrasion particles are easily identifiable with electron microscopy techniques by their morphology (sharp edges), size and chemical composition. Based on our longtime experience with tropospheric particles we can certainly exclude the carbonaceous particles to be abrasion products. In order to make this fact clear for the reader, we changed the paragraph to: "Besides C, the refractory carbonaceous particles always contain O and Si (Figures 2, 4 and 5), and in most cases also S. The element Si may at least partly be an artifact of the substrate. The S X-ray peak in EDX-spectra originates either from sulfates internally mixed with the carbonaceous particles or from stray radiation. Please note that the heights of the individual peaks in figure 2 are not proportional to the element concentrations, but give a rough estimate of the element abundance. The elements Cr, Fe, and Ni are often found as minor component (Figure 4). These three elements exclusively occur within the carbonaceous matrix, and are not abrasion products from ice particles hitting the aerosol inlet as the metallic particles described by Murphy et al. (2004) and Martinsson et al. (2014). Furthermore, none of the samples was collected during the existence of ice particles which could potentially remove material from the impactors' inlet. During collection of samples A, B, E and G, polar stratospheric cloud particles (PSC) containing oxides of nitrogen, NOy, were abundant. As we found the refractory carbonaceous particles in all samples independent of the occurrence of NOy, they are not artifacts from the removal of material from the inlet system."

- p. 7 1. 24: For how many particles was this element distribution imagine done? Fig. 6 shows just one. Is there any statistics on the results of this analysis?

Element distribution imaging and spot measurements were only conducted on four particles as this is a very time consuming approach. We did this in order to get to know if STEM shows any heterogeneous inclusions which we were not able to see either

with SEM (in backscattered imaging) or TEM. This little investigation shows, that the minor elements are homogeneously distributed within the particles, e.g. the particles do NOT contain elements heterogeneously distributed.

- p. 7 1. 25: I do not know how the element distribution images work, hence I do not know what "measuring several" (how many?) "points on the particles" means. Please explain this more in detail.

In element distribution images, the chemical composition of a single pixel within the image is reported. The images shown in fFig. 5 do all have a pixel size of 256x256. We changed the paragraph to: "The spatial distribution of minor elements within the carbonaceous particles was investigated by element distribution images in STEM (Figure 5) with a 256x256 pixel resolution as well as by measuring several points on the same particle. With both approaches it is possible to obtain highly resolved information on the spatial distribution of elements within a nanometer-scale particle. C is the most abundant element and is found in the whole particle."

- p. 7 1. 31: Why did you generate these four groups? If I did not overlook it, they are not used afterwards.

As we were aiming to characterize the refractory carbonaceous particles in as many details as possible, we found it useful to show the occurrence of the minor elements in the different samples. That some particles contain all of the minor elements Cr, Fe and Ni, but others do not contain any of them or just some is also shown in EDX-spectra in figure 2. In order to show the abundance of particles with or without minor elements, we decided to show this also in Table 2.

- p. 9 1. 22: The differences between your study and the results in Nguyen et al., 2008 is likely due to the different atmospheric measurement regions and different measurement altitudes. You should mention that, otherwise the reader might take the Nguyen reference as a contradiction to your findings, which is, in my point of view, not the case.

We agree that there are similarities to the results of Nguyen et al., 2008. Therefore we changed the paragraph to: "Mixed carbon-sulfur particles were observed by Nguyen et al. (2008) (diameter $\leq$ 1 $\mu$m) at 10 km altitude between 50°N and 30°S. These particles were assumed to have formed from condensed organic matter. The differences between these particles and those found in the current study might result from differences in sampling altitudes and regions. Therefore we cannot totally exclude the particles to be different, taking into account that the particles might have evolved from condensed organic matter. However, we do not know if secondary organic particles become refractory as a result of atmospheric processes."

- p. 10 1. 11: Pyro-convection is defined as fire-started or fire-added convection, hence the definition given by you is incomplete.

We have changed the sentence to "This material was thought to be injected into the stratosphere by the pyro-convective effect (i.e., fire-started or fire added convection)."

- p. 11 aircraft exhaust section: The Mazaheri et al. reference here, and later on also the Tumolva et al. and Torvela et al. references in the wood burning section, here you compare freshly emitted particle properties to your particles, which are, because they were measured in the polar vortex, likely more than one year old. This comparison can only fail, the particles aged and strongly changed. I miss this time effect in all potential source paragraphs, but this point is important and must be considered in the discussion section.

We agree and added the following paragraph before the "aircraft exhaust" section which regards all following paragraphs: "Most particle groups discussed in the following were collected close to their emission source. We are aware of the fact, that particles collected in the polar stratosphere may in principle change their properties during their atmospheric lifetime. However, ordering of carbonaceous material is an irreversible process leading always to a higher degree of ordering (Diessel et al., 1978; Itaya, 1981; Pesquera and Velasco, 1988). As most of the particles analyzed show no or

only very little ordering, it is assumed that the particles did not change their nanostructure during their atmospheric lifetime. On the other hand, several electron microscopy studies describe soot particles in the stratosphere (Pueschel et al., 1992, 1997; Sheridan et al., 1994; Strawa et al., 1999; Ebert et al., 2016). Thus, it can be expected that soot particles - once injected into the stratosphere – do not change their typical nanostructure under stratospheric conditions."

- p. 11 1. 26: Consider to add "(dominant meteorite fraction)" or something similar after "chondrites", in order to explain what this thing is.

We have changed the sentence to: "Carbonaceous material is observed in chondrites (dominant meteorite fraction) as well as in interplanetary dust particles (IDPs)."

- p. 15 1. 1: The summary is too short. You did a lot of work, please expand the summary.

We changed the summary to: "The major finding of the present study is that the refractory component consists of carbonaceous particles only, with a number mixing ratio of 1.1 (mg air)-1 (median for all samples). Most carbonaceous particles are not internally mixed with or coated by sulfates. The particles were sampled in air having low abundance of N2O and therefore long residence times in the stratosphere. Thus, one would expect them to be covered with condensed sulfuric acid resulting from the oxidation of COS (Wilson et al., 2008). The reason for this discrepancy is not known. As major elements only C and O were detected. Most of the carbonaceous particles show small and variable amounts of Fe, Cr and Ni. These minor elements are distributed in the carbonaceous matrix, i.e., they do not occur as heterogeneous inclusions. Most carbonaceous particles are completely amorphous. The exact source of the refractory carbonaceous particles remains unclear and can only be confined by exclusion. Based on the investigated physical properties and chemical composition of the particles, aircraft exhaust, volcanic emissions and biomass burning can be certainly excluded as source. The same is true for the even more unlikely sources wood burning, coal burning, diesel engines and ship emissions. It is expected that exhaust of rockets powered by kerosene or other hydrocarbons emit soot, but due to the lack of available electron microscopy studies of these emissions, rocket exhaust cannot be excluded as a possible source of the refractory carbonaceous particles found. Carbonaceous material from IDPs and extraterrestrial particles, likely originating from meteoric ablation and fragmentation remain as the most probable source for the particles encountered in the current study." We also deleted the last sentence as we have changed our conclusions based on the remarks of both reviewers and the interactive comment of Alexander D. James.

Technical corrections:

- p. 1 l. 29: Please remove the empty line, the last sentence of the abstract belongs to the upstream paragraph and should not be separated. Changed accordingly

- p. 2 l. 3: "sulfur" is an "element", not a "component". Changed accordingly

- p. 2 l. 13: which "groups" where identified? "Particle" or "morphology" or . . .

We have changed the sentence to: "However, due to the lack of instrumentation, the chemistry of the particles could not be investigated. Refractory particles with diameters >1 $\mu$m were studied in more detail by Zolensky and Mackinnon (1985), and several particle groups were distinguished. . ."

- p. 2 l. 15: "a large refractory particle load", what does this mean? With respect to particle mass or particle number or just fraction of particles containing refractory material?

We have changed the sentence to: "In contrast to prior findings, a large number of refractory stratospheric particles was recognized by Zolensky et al. (1989)"

- p. 2 l. 20: "widely distributed", what does this mean? All over the globe? Or at all altitudes (which ones?) in the area of investigation (which was?)?

We have changed the sentence to: "Carbonaceous aerosol was found to contribute to the aerosol population at all latitudes in the stratosphere and interplanetary dust was significantly abundant above 30 km for particles $\geq 0.35$ $\mu$m (Renard et al., 2008)."

- p. 2 1. 22: I'm not a native speaker, but shouldn't it be "Earth's"?

We have changed it accordingly

- p. 2 1. 31: "condensation of saturated gases", it is not necessary to provide seven (!) references for this textbook process. As it disturbs reading the paper, you should reduce the number.

We agree and deleted four of the references!

- p. 3 1. 25: Please insert a comma after "impactor".

Comma inserted.

- p. 3 1. 26: Please remove the "The" before "MACS".

"The" removed.

- p. 3 1. 32: "It was weaker . . ." What is "it"? The "Arctic winter"? But then the sentence does not make sense.

We have changed the sentence to "The vortex was weaker than the early winter polar vortices of the previous years."

- p. 4 1. 4: Please use "$\Theta$" instead of "PT".

Changed accordingly

- p. 4 1. 20: Please move "software" before the brackets.

Changed accordingly

- p. 4 sect. 2.2: Please use "EDX" instead of "energy-dispersive X-ray" throughoutly, after you defined it once.

Changed accordingly

- p. 6 1. 29: The whole statistical analysis section reads like a bullet point list. Please make it more a coherent text or a real bullet point list, with an introductory text.

We have changed the section to "Censored boxplots show data taking into account the fraction of values below detection limit. Lower and upper quartiles appear as a box, minimum and maximum values as whiskers. The differences in element ratios between samples collected inside and outside the vortex were tested with the generalized Wilcoxon test (Helsel, 2012) applying a significance-level of 5%. Furthermore, the differences in size, projected area diameter and element ratios between the various samples were tested with the Kruskal-Wallis rank sum test (uncensored data) and the generalized Wilcoxon test (censored data). In all individual tests, a significance level of 5 % was applied. The detection limits for EDX data were calculated from counting statistics (background counts plus three times standard deviation of background counts). All statistical calculations were performed with R (version 3.3.0; R Core Team, 2016) and using the contributed package NADA (version 1.5-6; Lee, 2013)."

- p. 6 1. 7: Please remove "applying a significance level of 5%", this is redundant, as it is mentioned in the last sentence of this paragraph.

Accordingly removed

- p. 6 1. 15: Please move the comma after "(Figure 1)".

Changed accordingly

- p. 7 1. 1: Please use "indicated" instead of "shown", you do not show real particle size distributions, e.g. dN/dlogDp.

Changed accordingly

- p. 7 1. 6: Please move "besides C" to the beginning of the sentence.

Changed accordingly

- p. 7 1. 25: Please replace "contained in the whole" with "found everywhere in".

Changed to "...and is found in the whole particle"

- p. 8 1. 18: Please replace "The samples" with "All samples".

Changed accordingly

- p. 10 1. 27: Please replace "emissions" with "eruptions".

Changed accordingly

- p. 11 1. 15: A space is missing before "The".

Changed accordingly

- p. 11 1. 16: Please replace "at" with "in".

Changed accordingly

- p. 12 1. 27: "comprised ... to" sounds strange, better use "contribute ...to" or something similar.

Changed accordingly

- Fig. 2: Please specify K$\alpha$ and K$\beta$ in the figure caption. What does "all particles" mean? In the stratosphere or all sample or all refractory? Is the peak height/area linearly representative for the number of atoms? This should be mentioned somewhere in the text.

We did the specifications as suggested. The figure caption now reads like: "Figure 2: TEM bright field image (a) of a typical refractory carbonaceous particle from sample H (19.1 km altitude). The image is representative for all refractory carbonaceous particles. The morphology is not depending on chemical composition, size, morphology or nanostructure. Energy-dispersive X-ray spectra of (b) a typical refractory carbonaceous particle with Fe, Cr and Ni, (c) Fe and Cr, (d) Fe and (e) without any other minor constitute. The particle predominantly consists of C and O. Minor amounts of Si are

always present and may partly be an artifact of the substrate. Cu is an artifact from the TEM grid. K$\alpha$ and K$\beta$ as well as L$\alpha$ and L$\beta$ denote different X-ray peaks emitted from the same element." EDX is a quantitative method. The height of individual peaks give a good estimate of the element abundance but is not linearly comparable because of different amounts of energy necessary to excite the individual elements; therefore every spectrum needs to be corrected in order to obtain quantitative amounts of elements. We introduced the following sentence to p.7 l.6 of the discussion paper: "Besides C, the refractory carbonaceous particles always contain O and Si (Figures 2, 4 and 5), and in most cases also S. The element Si may at least partly be an artifact of the substrate. The S X-ray peak in EDX-spectra originates either from sulfates internally mixed with the carbonaceous particles or from stray radiation. Please note that the heights of the individual peaks in figure 2 are not proportional to the element concentrations, but give a rough estimate of the element abundance."

- Fig. 3: The given particle numbers are the total number of analyzed particles or only the refractory ones? Please specify this "n" in the figure caption.

We removed this figure from the manuscript for reasons given above.

- Fig. 6: The colors in the lower row of pictures are hard to see. I believe to use bright red or even white as occurrence indicator color in all pictures would improve the figure.

We agree. We decided to choose black/white.

---

## Author Comment (AC3) · 30 Aug 2017

We gratefully acknowledge the suggestions of Alexander D. James and included them to the revised version of the paper. We believe that the changes considerably helped to improve the quality of the manuscript.

This article has the potential to be a useful addition to the understanding of stratospheric aerosol, particularly since there are relatively few capture and return samples with statistics on this level. Whilst the conclusions they are able to draw are limited, publication of such data is vital in facilitating future understanding. I feel that the authors have missed or omitted a section of the literature which, once considered, can both add to the understanding of the results and increase the potential audience of the

article. Below are a number of specific comments on language, formatting and content. The authors provide a good introduction to the current and historical field. I was surprised to see no reference to the recent and thorough review of Kremser et al. (2016), who summarised some of the studies mentioned in the introduction of this work and other related topics.

We have mentioned the publication of Kremser et al. (2016) lateron at the beginning of chapter 3. But we agree that the mentioned paper is a very good recent summary of past findings and therefore decided to add the following sentence to the end of the first paragraph of the introduction: "A comprehensive summary of stratospheric aerosol and sulfur chemistry is given by Kremser et al. (2016)."

Page 4 line 2; change "extend" to read "extent".

Accordingly changed.

The analysis of images for structure of carbonaceous material is interesting. Was electron diffraction data recorded for any samples?

Unfortunately it was not possible to conduct electron diffraction with the samples.

Page 5 line 8; reformat $5 \times 10\text{-}3$.

Accordingly changed.

Page 5 line 27; amend to "too close to"

Accordingly changed.

Page 5 line 30 onwards; sentence is hard to understand. Perhaps "Any particle which showed no signs of destruction or morphological change was taken to be non-volatile. Particles which changed under the electron beam were deemed volatile, allowing quantification of the fraction of aerosol which is volatile."

We have removed the whole paragraph.

Section 4.1; I believe this section would benefit from also discussing the size ranges of the various particles. For example Ebert et al. (2016) discuss mainly particles of radius greater than 500 nm, which have metallic or meteoritic composition. In that study the smaller size fraction is described as being largely carbonaceous material in sulfate liquid droplets, similar to the findings of this study.

We added the size ranges of the particles described in that section in arrows: Blake and Kato (1995): $\leq$ 0.5 nm; Pueschel et al. (1992): $\sim$0.2 – 0.3 $\mu$m; Pueschel et al. (1997): $\leq$ 1um; Sheridan et al. (1994): $\sim$ 0.3 $\mu$m; Strawa et al. (1999): $\sim$0.3 - 0.4 $\mu$m; Chuan and Woods (1984): $\sim$0.1 $\mu$m; Ebert et al. (2016): $\leq$ 0.500 $\mu$m; Chen et al. (1998): $\leq$ 0.1 - 2 $\mu$m; Nguyen et al., 2008: $\leq$ 1 $\mu$m; Baumgardner et al. (2004): 0.15 - 1 $\mu$m; Schwarz et al. (2006): 0.15 - 0.7 $\mu$m; Renard et al. (2008): 0.35 - 2 $\mu$m

Page 12 line 1; change to "the particles described above matches the refractory..."

Accordingly changed.

Possibility that particles have an extraterrestrial origin; this section makes a good comparison between measurements of extraterrestrial material and the particles observed in the stratosphere. What is lacking is any discussion of the process which occur as a result of frictional heating during atmospheric entry. There is currently some discussion of whether unablated meteoric material will sediment too rapidly to be found in the stratosphere (Carrillo-Sánchez et al., 2016), or whether significant fragmentation of ablating meteorites could lead to smaller aerosol with longer atmospheric lifetimes (Subasinghe et al., 2016). Considering these processes in the light of the results presented here would broaden the appeal of the current results to a wider audience and add significantly to the conclusions the authors are able to draw from their data. Regarding ablation: The fact that the three metals discussed; Ni, Fe and Cr; have ratios significantly different than their chondritic abundances has rather more interesting implications when considered with respect to the ablation process. Since in interplanetary dust Ni is largely contained in relatively volatile metal phases (melting points

around 1200-1500 K), Fe is spread between volatile metallic and more refractory silicate phases (melting point >1800 K) and Cr is contained in the less volatile silicates (Bunch and Olsen, 1975), the three elements will ablate rather differently (Gómez-Martín et al., 2017). The relative volatility of Ni is therefore reconcilable with the larger Ni/Fe ratio measured here and does not rule out an extraterrestrial source, but the high Cr/Fe and Cr/Ni suggest that Cr at least has a terrestrial source, since if anything Cr should ablate less completely than the other elements. Regarding fragmentation: This is hypothesized to happen by evaporation of volatile phases such as iron sulfides and amorphous carbonaceous material (ordered graphitic material would be much more refractory). It may be reasonable as a result that the metal bearing silicates would remain in larger particles which have very short lifetimes in the stratosphere, but carbonaceous material and some additional Fe is atmospherically available as a result. The question could possibly be more constructively phrased in another way. Since we know that meteoric ablation occurs and meteoric smoke forms, why is it not unequivocally observed in these measurements? Indeed numerical modelling of MSPs suggests that they should be observable in this size range (Bardeen et al., 2008). Could it be that nucleation, growth and sedimentation of crystalline PSC has removed meteoric material? What implications would the partial dissolution of metals have on these measurements? Could dissolution, precipitation and agglomeration in liquid droplets cause more rapid growth of MSP compared to model predictions?

We highly appreciate this comment! Considering the given comment and included literature, we included the suggestions into the paper. Now, we added the following paragraph to the "extraterrestrial particles" section in 4.2: "The chemical composition of extraterrestrial material may be strongly fractionated by frictional heating during atmospheric entry (e.g., Carrillo-Sánchez et al., 2016; Gómez- Martin et al., 2017). The processes taking place during atmospheric entry include ablation by sputtering and thermal evaporation as well as fragmentation. Meteorite ablation particles usually occur as iron, glass or silicate spheres (e.g., Blanchard et al., 1980; Murrell et al., 1980). Submicrometer refractory carbonaceous particles resulting from meteoric ablation and

fragmentation have - to the best of our knowledge - not been described in previous literature. However, it is conceivable that such particles originate from carbonaceous material contained in meteorites or interplanetary dust particles. The observed non-chondritic ratios of the minor elements Fe, Cr, Ni are not a strong argument against such an origin, as the volatility of these elements depends on the minerals in which they are contained. Most of extraterrestrial Fe occurs as metal, silicate or oxide, most of Ni as metal (Papike, 1998). Cr may occur as oxide, sulphide or nitride as well as a minor component in metal and silicates (Bunch and Olsen, 1975). Depending on the relative abundance of the different mineral phases, substantial fractionation during evaporation can be expected (see also Floss et al., 1996). In summary, meteoric ablation and fragmentation particles are a possible source of the particles encountered in the present study." In addition, we added to the abstract: "Recondensed organic matter and extraterrestrial particles, potentially originating from ablation and fragmentation remain as possible sources of the refractory carbonaceous particles studied. However, additional work is required in order to identify the sources unequivocally.", and to the summary: "Carbonaceous material from IDPs and extraterrestrial particles, likely originating from meteoric ablation and fragmentation remain as the most probable source for the particles encountered in the current study." These comments also inherently include the issue of sample aging, which both anonymous reviewers rightly mention. Having some experience of electron microscopy, I suspect that this statistical detail could only be reached from measurements which would take several years to make. In addition to the reviewer's comments then, the possibility should be considered that some samples have aged more than others.

According to this comment we added the following sentence at the end of chapter 2.1: "Anyhow, it should be kept in mind that other parameters (chemical composition, mixing state) may be changed to a variable extent by aging."

References Bardeen, C. G., Toon, O. B., Jensen, E. J., Marsh, D. R., and Harvey, V. L.: Numerical simulations of the three-dimensional distribution of meteoric dust in

the mesosphere and upper stratosphere, J. Geophys. Res.: Atmos., 113, D17202, 2008. Bunch, T. E., and Olsen, E.: Distribution and significance of chromium in meteorites, Geochim. Cosmochim. Acta, 39, 911-927, http://dx.doi.org/10.1016/0016-7037(75)90037-X, 1975. Carrillo-Sánchez, J. D., Nesvorná, D., Pokorná, P., Janches, D., and Plane, J. M. C.: Sources of cosmic dust in the Earth's atmosphere, Geophys. Res. Lett., 43, 11,979- 911,986, 10.1002/2016GL071697, 2016. Ebert, M., Weigel, R., Kandler, K., Günther, G., Molleker, S., Grooß, J. U., Vogel, B., Weinbruch, S., and Borrmann, S.: Chemical analysis of refractory stratospheric aerosol particles collected within the arctic vortex and inside polar stratospheric clouds, Atmos. Chem. Phys., 16, 8405-8421, 2016. Gómez-Martín, J. C., Bones, D. L., Carrillo-Sánchez, J. D., James, A. D., Trigo- Rodríguez, J. M., B. Fegley, J., and Plane, J. M. C.: Novel experimental simulations of the atmospheric injection of meteoric metals, Astrophys. J., 836, 212, 2017. Kremser, S., Thomason, L. W., von Hobe, M., Hermann, M., Deshler, T., Timmreck, C., Toohey, M., Stenke, A., Schwarz, J. P., Weigel, R., Fueglistaler, S., Prata, F. J., Vernier, J. P., Schlager, H., Barnes, J. E., Antuña-Marrero, J.-C., Fairlie, D., Palm, M., Mahieu, E., Notholt, J., Rex, M., Bingen, C., Vanhellemont, F., Bourassa, A., Plane, J. M. C., Klocke, D., Carn, S. A., Clarisse, L., Trickl, T., Neely, R., James, A. D., Rieger, L., Wilson, J. C., and Meland, B.: Stratospheric aerosol—Observations, processes, and impact on climate, Rev. Geophys., 54, 278-335, 2016. Subasinghe, D., Campbell-Brown, M. D., and Stokan, E.: Physical characteristics of faint meteors by light curve and high-resolution observations, and the implications for parent bodies, Mon. Not. Royal Astro. Soc., 457, 1289-1298, 2016.